# Fractal symmetric phases of matter

**Trithep Devakul[1][*], Yizhi You[2], F. J. Burnell[3] and S. L. Sondhi[1]**

**1** Department of Physics, Princeton University, NJ 08544, USA.
**2** Princeton Center for Theoretical Science, Princeton University, NJ 08544, USA.
**3** Department of Physics, University of Minnesota Twin Cities, MN 55455, USA.

[*] tdevakul@princeton.edu

## Abstract

We study spin systems which exhibit symmetries that act on a fractal subset of sites, with fractal structures generated by linear cellular automata. In addition to the trivial symmetric paramagnet and spontaneously symmetry broken phases, we construct additional fractal symmetry protected topological (FSPT) phases via a decorated defect approach. Such phases have edges along which fractal symmetries are realized projectively, leading to a symmetry protected degeneracy along the edge. Isolated excitations above the ground state are symmetry protected fractons, which cannot be moved without breaking the symmetry. In 3D, our construction leads additionally to FSPT phases protected by higher form fractal symmetries and fracton topologically ordered phases enriched by the additional fractal symmetries.

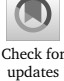

# 1  Introduction

Symmetries are indispensable in the characterization and classification of phases of matter. In many cases, knowledge of the systems symmetries and how they are respected or spontaneously broken provide a complete description of a phase. Beyond the usual picture of spontaneously broken symmetries, it has been recently appreciated that multiple phases with the same unbroken symmetry can also exist, known as symmetry protected topological (SPT) phases, which has been the subject of great interest in a variety of systems [1–23]. These phases cannot be connected adiabatically while maintaining the symmetry, but *can* be so connected if the symmetry is allowed to be broken.

The vast majority of these cases deal with *global* symmetries, symmetries whose operation acts on an extensive volume of the system. On the opposite end of the spectrum, one also has systems with emergent local *gauge* symmetries [24], which act on a strictly local finite part of the system; such symmetries may lead to topologically ordered phases [25, 26]. In between these two, one also has symmetries which act on some subextensive $d$-dimensional (integer $d$) *subsystem* of a system, such as along planes or lines in a 3D system; these are intermediate between gauge and global symmetries, and have as such also been called gauge-like symmetries. Systems with such symmetries display interesting properties [27–29], and are intimately related to models of *fracton topological order* [30–35] through a generalized gauging procedure [36, 37] (Type-I fracton order in the classification of Ref 37). Fracton topological order is a novel type of topological order, characterized by subextensive topology-dependent ground state degeneracy and immobile quasiparticle excitations, and has inspired much recent research [35–62]. In a recent work by the present authors, such subsystem symmetries were shown to also lead to new phases of matter protected by the collection of such symmetries [63].

The subject of this paper is yet another type of symmetry, which may be thought of as being "in between" two of the aforementioned subsystem symmetries: fractal symmetries. These act on a subset of sites whose volume scales with linear size $L$ as $L^d$ with some fractal dimension $d$ that is in general not integer. Note that these models have symmetries which act on a fractal

subset of a regular lattice, and should be distinguished from models (with possibly global symmetries) on fractal lattices [64–69]. Systems with such symmetries appear most notably in the context of glassiness in translationally invariant systems [34], such as the Newman-Moore model [70–78]. Via the gauge duality [36, 37], systems with such symmetries may describe theories with (Type-II) fracton topological order [32, 33]; these have ground state degeneracies on a 3-torus that are complicated functions of the system size, and immobile fracton excitations which appear at corners of *fractal* operators. Indeed, the recent excitement in the study of fracton phases arose from the discovery of the Type-II fracton phase exemplified by Haah's cubic code [33].

We focus on a class of fractal structures on the lattice that are generated by cellular automata (CA), from which many rich structures emerge [79–83]. In particular, we will focus on CA with linear update rules, from which self-similar fractal structures are guaranteed to emerge, following Ref 32. We construct a number of spin models which are symmetric under operations that involve flipping spins along these fractal structures. Unlike a global symmetry, the order of the total symmetry group may scale exponentially with system size, and therefore their case is more like that of subsystem symmetries.

We first present in Sec 2 a brief introduction to CA, and how fractal structures emerge naturally from them. When dealing with such fractals, a polynomial representation makes dealing with seemingly complicated fractal structures effortlessly tractable (see Ref [32]), and we encourage readers to become familiar with the notation. In Sec 3, we take these fractal structures to define symmetries on a lattice in 2D. These symmetries are most naturally defined on a semi-infinite lattice; here, symmetries flip spins along fractal structures (e.g. translations of the Sierpinski triangle). We describe in detail how such symmetries should be defined on various other lattice topologies, including the infinite plane. Simple Ising models obeying these symmetries are constructed in Sec 4, which demonstrate a spontaneously fractal symmetry broken phase at zero temperature, and undergoes a quantum phase transition to a trivial paramagnetic phase.

In Sec 5 we use a decorated defect approach to construct fractal SPT (FSPT) phases. The nontriviality of these phases are probed by symmetry twisting experiments and the existence of symmetry protected ungappable degeneracies along the edge, due to a locally projective representation of the symmetries. Such phases have symmetry protected *fracton* excitations that are immobile and cannot be moved without breaking the symmetries or creating additional excitations.

Finally, we discuss 3D extensions in Sec 6, these include models similar to the 2D models discussed earlier, but also novel FSPT phases protected by a combination of regular fractal symmetries and a set of symmetries which are analogous to *higher form* fractal symmetries. These FSPT models with higher form fractal symmetries, in one limit, transition into a fracton topologically ordered phase while still maintaining the fractal symmetry. Such a phase describes a topologically ordered phase enriched by the fractal symmetry, thus resulting in a fractal symmetry enriched (fracton) topologically ordered (fractal SET [84–92], or FSET) phase.

## 2   Cellular Automata Generate Fractals

We first set the stage with a brief introduction to a class of one-dimensional CA, from which it is well known that a wide variety of self-similar *fractal* structures emerge. In latter sections, these fractal structures will define *symmetries* which we will demand of Hamiltonians.

Consider sites along a one-dimensional chain or ring, each site $i$ associated with a $p$-state variable $a_i \in \{0, 1, \ldots, p-1\}$ taken to be elements of the finite field $\mathbb{F}_p$. We define the *state* of

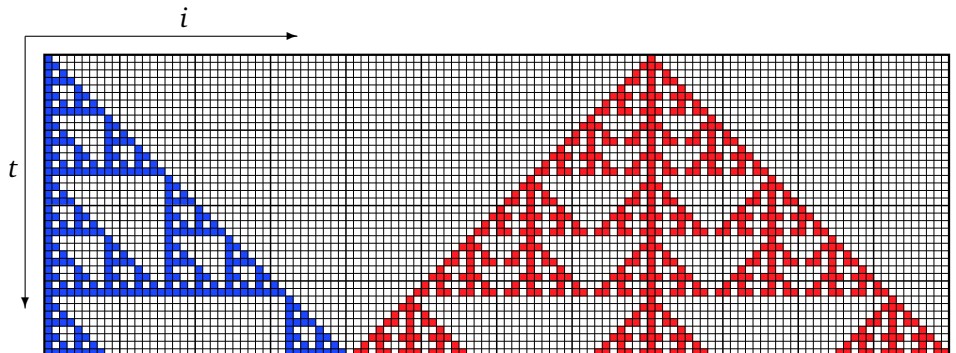

Figure 1: Fractal structures generated by (left,blue) the Sierpinski rule $a_i^{(t+1)} = a_{i-1}^{(t)} + a_i^{(t)}$ and (right,red) the Fibonacci rule $a_i^{(t+1)} = a_{i-1}^{(t)} + a_i^{(t)} + a_{i+1}^{(t)}$, starting from the initial state $a_i^{(0)} = \delta_{i,0}$. In the polynomial representation, the row $t$ is given by $f(x)^t$, with (blue) $f(x) = 1 + x$ and (red) $f(x) = x^{-1} + 1 + x$ over $\mathbb{F}_2$. Notice that self-similarity at every row $t = 2^l$ (here, we show evolution up to $t = 40$).

the CA at time $t$ as the set of $a_i^{(t)}$. We will typically take $p = 2$, although our discussion may be easily generalized to higher primes. We consider CA defined by a set of translationally-invariant local *linear update rules* which determine the state $\{a_i^{(t+1)}\}$ given the state at the previous time $\{a_i^{(t)}\}$. Linearity here means that the future state of the $i$th cell, $a_i^{(t+1)}$, may be written as a linear sum of $a_j^{(t)}$ for $j$ within some small local neighborhood of $i$. Throughout this paper, all such arithmetic is integer arithmetic modulo $p$, following the algebraic structure of $\mathbb{F}_p$. Figure 1 shows two sets of linear rules which we will often refer to:

1. The Sierpinski rule, given by $a_i^{(t+1)} = a_{i-1}^{(t)} + a_i^{(t)}$ with $p = 2$, so called because starting from the state $a_i^{(0)} = \delta_{i,0}$ one obtains Pascal's triangle modulo 2, who's nonzero elements generate the Sierpinski triangle with fractal Hausdorff dimension $d = \ln 3 / \ln 2 \approx 1.58$. In the polynomial representation (to be introduced shortly), this rule is given by $f(x) = 1 + x$.

2. The Fibonacci rule, $a_i^{(t+1)} = a_{i-1}^{(t)} + a_i^{(t)} + a_{i+1}^{(t)}$ also with $p = 2$, so called because starting from $a_i^{(0)} = \delta_{i,0}$ it generates a fractal structure with Hausdorff dimension $d = 1 + \log_2(\varphi) \approx 1.69$ with $\varphi$ the golden mean [32]. The polynomial representation is given by $f(x) = x^{-1} + 1 + x$.

Fractal dimensions for CA with linear update rules may be computed efficiently [95].

To see why such linear update rules always generate self-similar structures, it is convenient to pass to a polynomial representation. We may represent the state $a_i^{(t)}$ as a Laurent polynomial $s_t(x)$ over $\mathbb{F}_p$ as

$$s_t(x) = \sum_{i=-\infty}^{\infty} a_i^{(t)} x^i \tag{1}$$

for an infinite chain. Alternatively, periodic boundary conditions may be enforced by setting $x^L = 1$. In this language, these update rules take the form

$$s_{t+1}(x) = f(x) s_t(x) \tag{2}$$

for some polynomial $f(x)$ containing only small finite powers (both positive or negative) of $x$. For the Sierpinski rule we have $f(x) = 1 + x$, and for the Fibonacci rule we have $f(x) = x^{-1} + 1 + x$.

Then, given an initial state $s_0(x)$, we have that

$$s_t(x) = f(x)^t s_0(x). \tag{3}$$

A neat fact about polynomials in $\mathbb{F}_p$ is that they obey what is known as the *Freshman's Dream*,

$$f(x) = \sum_i c_i x^i \implies f(x)^{p^k} = \sum_i c_i x^{ip^k}, \tag{4}$$

whenever $t$ is a power of $p$. This can be shown straightforwardly by noting that the binomial coefficient $\binom{p^k}{n}$ is always divisible by $p$ unless $n = 0$ or $n = p^k$.

It thus follows that such CA generate fractal structures. Let us illustrate for the Fibonacci rule starting from the initial configuration $s_0(x) = 1$, i.e. the state where all $a_i = 0$ except for $a_0 = 1$. Looking at time $t = 2^l$, the state is $s_t(x) = x^{-2^l} + 1 + x^{2^l}$. In the following evolution, each of the non-zero cells $a_{-2^l} = a_0 = a_{2^l} = 1$ each look locally like the initial configuration $s_0$, and thus the consequent evolution results in three shifted structures identical to the initial evolution of $s_0$ (up until they interfere), as can be seen in Figure 1. At time $t = 2^{k+1}$, this process repeats but at a larger scale. Thus, we can see that any linear update rule of this kind will result in self-similar fractal structures when given the initial state $s_0(x) = 1$. As the rules are linear, all valid configurations correspond to superpositions of this shifted fractal.

The entire time evolution of the CA may be described at once by a single polynomial $F(x, y)$ over two variables $x$ and $y$,

$$F(x, y) = \sum_{t=0}^{\infty} f(x)^t y^t, \tag{5}$$

and we have that the coefficient of $y^t$ in $F(x, y)s_0(x)$ is exactly $s_t(x) = f(x)^t s_0(x)$.

The two-dimensional fractal structures in Figure 1 generated by these CA emerge naturally due to a set of simple local constraints given by the update rules. In the next section, we will describe 2D classical spin Hamiltonians which energetically enforce these local constraints. The ground state manifold of these classical models is described exactly by a valid CA evolution, which we will then take to define *symmetries*.

## 3   Fractal Symmetries

To discuss physical spin Hamiltonians and symmetries, it is useful to also use a polynomial representation of operators. Such polynomial representations are commonly used in classical coding theory [96], and refined in the context of translationally invariant commuting projector Hamiltonians by Haah [97]. We will utilize only the basic tools (following much of Ref [32]), and specialize to Pauli operators ($p = 2$ from the previous discussion), although a generalization to $p$-state Potts spins is straightforward.

Let us consider in 2D a square lattice with one qubit (spin-1/2) degree of freedom per unit cell. Acting on the qubit at site $(i, j) \in \mathbb{Z}^2$, we have the three anticommuting Pauli matrices $\hat{Z}_{ij}$, $\hat{X}_{ij}$, and $\hat{Y}_{ij}$. We define the function $Z(\cdot)$ from polynomials in $x$ and $y$ over $\mathbb{F}_2$ to products of Pauli operators, such that acting on an arbitrary polynomial we have

$$Z\left(\sum_{ij} c_{ij} x^i y^j\right) = \prod_{ij}(\hat{Z}_{ij})^{c_{ij}}, \tag{6}$$

and similarly for $X(\cdot)$ and $Y(\cdot)$. For example, we have $Z(1 + x + xy) = Z_{0,0}Z_{1,0}Z_{1,1}$. Some useful properties are that the product of two operators is given by the sum of the two polynomials, $Z(\alpha)Z(\beta) = Z(\alpha + \beta)$, and a translation of $Z(\alpha)$ by $(i, j)$ is given by $Z(x^i y^j \alpha)$.

Perhaps the most useful property of this notation is that two operators $Z(\alpha)$ and $X(\beta)$ anticommute if and only if $[\alpha\bar{\beta}]_{x^0 y^0} = 1$, where $[\cdot]_{x^i y^j}$ denotes the coefficient of $x^i y^j$ in the polynomial, and we have introduced the *dual*,

$$p(x, y) = \sum_{ij} c_{ij} x^i y^j \leftrightarrow \bar{p}(x, y) = \sum_{ij} c_{ij} x^{-i} y^{-j}, \tag{7}$$

which may be thought of as the spatial inversion about the point $(0, 0)$. We will also often use $\bar{x}$ to represent $x^{-1}$ for convenience. More usefully, we may express the commutation relation between $Z(\alpha)$ and translations of $X(\beta)$ (given by $X(x^i y^j \beta)$) as

$$Z(\alpha) X(x^i y^j \beta) = (-1)^{d_{ij}} X(x^i y^j \beta) Z(\alpha), \tag{8}$$

where $d_{ij}$ may be computed directly from the *commutation polynomial* of $\alpha$ and $\beta$,

$$P(\alpha, \beta) = \sum_{ij} d_{ij} x^i y^j = \alpha \bar{\beta}, \tag{9}$$

which may easily be computed directly given $\alpha$ and $\beta$. In particular, $P = 0$ would imply that every possible translations of the two operators commute.

## 3.1 Semi-infinite plane

We may now transfer our discussion of the previous section here. Let us consider a semi-infinite plane, such that we only have sites $(i, j)$ with $x^i y^{j \geq 0}$. We may then interpret the $j$th row as the state of a CA at time $j$, starting from some initial state at row $j = 0$. Consider the linear CA with update rule given by the polynomial $f(x)$, as defined in Eq 2. The classical Hamiltonian which energetically enforces the CA's update rules is given by

$$\mathcal{H}_{\text{classical}} = -\sum_{i=-\infty}^{\infty} \sum_{j=1}^{\infty} Z(x^i y^j [1 + \bar{f}\bar{y}]), \tag{10}$$

where we have excluded terms that aren't fully inside the system.

As an example, consider the Sierpinski rule $f = 1 + x$ ($f$ will always refer to a polynomial in only $x$). Equation 10 for this rule gives,

$$\mathcal{H}_{\text{Sierpinski}} = -\sum_{ij} \hat{Z}_{ij} \hat{Z}_{i, j-1} \hat{Z}_{i-1, j-1}, \tag{11}$$

which is exactly the Newman-Moore (NM) model originally of interest due to being an exactly-solvable translationally invariant model with glassy relaxation dynamics [70]. The NM model was originally described in a more natural way on the triangular lattice as the sum of three-body interactions on all *downwards facing* triangles, $\mathcal{H}_{\text{NM}} = -\sum_{\triangledown} ZZZ$. This model does not exhibit a thermodynamic phase transition (similar to the 1D Ising chain). Fractal codes based on higher-spin generalizations of this model have also been shown to saturate the theoretical information storage limit asymptotically [98].

We will be interested in the symmetries of such a model that involve flipping subsets of spins. Due to the deterministic nature of the CA, such operation must involve flipping some subset of spins on the first row, along with an appropriate set of spins on other rows such that the total configuration remains a valid CA evolution. Operationally, symmetry operations are given by various combinations of $F(x, y)$ (Eq 5). That is, for any polynomial $q(x)$, we have a symmetry element

$$S(q(x)) = X(q(x) F(x, y)). \tag{12}$$

Here, $q(x)$ has the interpretation of being an initial state $s_0$, and $S(q(x))$ flips spins on all the sites corresponding to the time evolution of $s_0$. As the update rules are linear, this operation always flips between valid CA evolutions. For example, $S(1)$ will correspond to flipping spins along the fractals shown in Fig 1.

To confirm that this symmetry indeed commutes with the Hamiltonian, we may use the previously discussed technology (Eq 8 and 9) to compute the commutation polynomial between $S(q(x))$ and translations of the Hamiltonian term $Z(1 + \bar{f}\bar{y})$,

$$
\begin{aligned}
P &= q(x)F(x,y)(1+fy) = q(x)(1+fy)\sum_{l=0}^{\infty}(fy)^l \\
&= q(x)\left(\sum_{l=0}^{\infty}(fy)^l + \sum_{l=1}^{\infty}(fy)^l\right) = q(x).
\end{aligned}
\tag{13}
$$

Since terms which have shift $y^0$ are not included in the Hamiltonian (Eq 10), this operator therefore fully commutes with the Hamiltonian. We may pick as a basis set of independent symmetry elements, $S(q_\alpha)$, for $\alpha \in \mathbb{Z}$ with $q_\alpha(x) = x^\alpha$. These operators correspond to flipping spins corresponding to the colored pixels in Fig 1, and horizontal shifts thereof. Each of these symmetries act on a fractal subset of sites, with volume scaling as the Hausdorff dimension of the resulting fractal.

## 3.2 Cylinder

Rather than a semi-infinite plane, let's consider making the $x$ direction periodic with period $L$, such that $x^L = 1$, while the $y$ direction is either semi-infinite or finite. In this case, there are a few interesting possibilities.

### 3.2.1 Reversible case

In the case that there exists some $\ell$ such that $f^\ell = 1$, then the CA is *reversible*. That is, for each state $s_t$, there exists a unique state $s_{t-1}$ such that $s_t = fs_{t-1}$, given by $s_{t-1} = f^{\ell-1}s_t$.

A proof of this is straightforward, suppose there exists two distinct previous states $s_{t-1}$, $s'_{t-1}$, such that $fs_{t-1} = fs'_{t-1} = s_t$. As they are distinct, $s_{t-1} + s'_{t-1} \neq 0$. However,

$$
0 = s_t + s_t = f(s_{t-1} + s'_{t-1}) = f^{\ell-1}f(s_{t-1} + s'_{t-1}) = s_{t-1} + s'_{t-1} \neq 0,
\tag{14}
$$

there is a contradiction. Hence, the state $s_{t-1}$ must be unique. The inverse statement, that a reversible CA must have some $\ell$ such that $f^\ell = 1$, is also true.

In this case, all non-trivial symmetries extend throughout the cylinder, and their patterns are periodic in space with period dividing $\ell$. An example of this is the Fibonacci rule with $L = 2^m$, for which $f^{L/2} = 1$. There are $L$ independent symmetries on either the infinite or semi-infinite cylinder, and the total symmetry group is simply $\mathbb{Z}_2^L$. The symmetries on an infinite cylinder are given by $S(q) = X\left(q(x)\sum_{l=-\infty}^{\infty}(fy)^l\right)$, where $f^{-1} \equiv f^{\ell-1}$.

### 3.2.2 Trivial case

If there exists $\ell$ such that $f^\ell = 0$, then the model is effectively trivial. All initial states $s_0$ will eventually flow to the trivial state $s_\ell = 0$. On a semi-infinite cylinder, possible "symmetries" will involve sites at the edge of the cylinder, but will not extend past $\ell$ into the bulk of the cylinder. On an infinite cylinder, there are no symmetries at all. An example of this is the Sierpinski rule with $L = 2^m$, for which $f^L = 0$.

### 3.2.3 Neither reversible nor trivial

If the CA on a cylinder is neither reversible nor trivial, then every initial state $s_0$ must eventually evolve into some periodic pattern, such that $s_t = s_{t+T}$ for some period $T$ at large enough $t$ (this follows from the fact that there are only finitely many states). Thus, there will be symmetry elements whose action extends throughout the cylinder, like in the reversible case. Interestingly, however, irreversibility also implies the existence of symmetry elements whose action is restricted only to the edge of the cylinder, much like the trivial case.

Let us take two distinct initial states $s_0, s'_0$ that eventually converge on to the same state at time $\ell$. Then, let $\tilde{s}_0 = s_0 + s'_0 \neq 0$ be another starting state. After time $\ell$, $\tilde{s}_\ell = s_\ell + s'_\ell = 0$, this state will have converged on to the trivial state. Thus, the symmetry element corresponding to the starting state $\tilde{s}_0$ will be restricted only to within a distance $\ell$ of the edge on a semi-infinite cylinder.

On an infinite cylinder, only the purely periodic symmetries will be allowed, so the total number of independent symmetry generators is reduced to between 0 and $L$.

## 3.3 On a torus

Let us next consider the case of an $L_x \times L_y$ torus. Symmetries on a torus must take the form of valid CA *cycles* on a ring of length $L_x$ with period $L_y$. The order of the total symmetry group is the total number of distinct cycles commensurate with the torus size, which in general does not admit a nice closed-form solution, but has been studied in Ref 94. Equivalently, there is a one-to-one correspondence between elements in the symmetry group and solutions to the equation

$$q(x)f(x)^{L_y} = q(x), \tag{15}$$

with $x^{L_x} = 1$. This may be expressed as a system of linear equations over $\mathbb{F}_2$, and can be solved efficiently using Gaussian elimination. For each solution $q(x)$ of the above equation, the action of the corresponding symmetry element is given by

$$S(q) = X\left(q(x)\sum_{l=0}^{L_y-1}(fy)^l\right). \tag{16}$$

As an example, consider the Sierpinski model on an $L \times L$ torus. Let $k(L) = \log_2(N_{\text{sym}}(L))$ be the number of independent symmetry generators, where $N_{\text{sym}}(L)$ is the order of the symmetry group. We are free to pick some set of $k(L)$ independent symmetry operators as a basis set (there is no most natural choice for basis), which we label by $q_\alpha(x)$ with $0 \leq \alpha < k$. To illustrate that $k(L)$ is in general a complicated function of $L$, we show in Table 2 $k(L)$ and a choice of $q_\alpha^{(L)}(x)$ for the few cases of $L$ where the number of cycles can be solved for exactly. An interesting point is that for the Sierpinski rule, $f(x)^{2^l} = 0$, thus for $L = 2^l$, there are no non-trivial solutions to Eq 15 and so $k(2^l) = 0$. To contrast, the Fibonacci rule has $f(x)^{2^l} = 1$, and so $k(L = 2^l) = L$.

## 3.4 Infinite plane

Now, let us consider defining such symmetries directly on an infinite plane, where we allow all $x^i y^j$. In the CA language, we are still free to pick the CA state at time, say $t = 0$, $s_0(x)$, which completely determines the CA states at times $t > 0$. However, we run into the issue of reversibility — how do we determine the history of the CA for times $t < 0$ which lead up to $s_0$? For general CA, there may be zero or multiple states $s_{-1}$ which lead to the same final state $s_0$. For a linear CA on an infinite plane, however, there is always at least one $s_{-1}$. We give an algorithm for picking out a particular history for $s_0$, and discuss the sense in which it

| $L$ | $k(L)$ | $q_\alpha^{(L)}(x)$ |
|---|---|---|
| $2^m$ | 0 | - |
| $2^m - 1$ | $L-1$ | $x^\alpha(1+x)$ |
| $2^m + 2^n - 1$ | $\gcd(L, 2^{n+1}-1)-1$ | $x^\alpha(1+x)\sum_{l=0}^{\frac{L}{k+1}-1}(x^{2^m-2^n})^l$ |
| $2m$ | $2k(m)$ | $x^{\alpha \bmod 2}[q_{\lfloor\alpha/2\rfloor}^{(m)}(x)]^2$ |

Figure 2: The number of independent symmetry generators $k(L)$ and a choice of $q_\alpha(x)$ for the Sierpinski model on an $L \times L$ torus for few particular $L$. Here, $m, n$ are positive integers, $m > n$, and $0 \le \alpha < k$ labels the symmetry polynomials $q_\alpha^{(L)}(x)$, and $\lfloor \cdot \rfloor$ denotes the floor function.

is a complete description of all possible symmetries, despite this reversibility issue. For this discussion it is convenient to, without loss of generality, assume $f$ contains at least a positive power of $x$ (we may always perform a coordinate transformation to get $f$ into such a form).

The basic idea is as follows: we have written the Hamiltonian (Eq 10) in a form that explicitly picks out a direction ($y$) to be interpreted as the time direction of the CA. However, we may always write the same term as a higher-order linear CA that propagates in the $\bar{x}$ direction,

$$1 + \bar{f}\bar{y} = \bar{x}^a\bar{y}\left[1 + \sum_{n=1}^{n_{max}} g_n(y)x^n\right] \equiv \bar{x}^a\bar{y}[1 + g(x,y)x], \tag{17}$$

where $a > 0$ is the highest power of $x$ in $f$, $n_{max}$ is finite, and $g_n(y)$ is a polynomial containing only non-negative powers of $y$. This describes an $n_{max}$-order linear CA. For the Sierpinski rule, we have only $g_1(y) = 1 + y$, and for the Fibonacci rule we have both $g_1(y) = 1 + y$ and $g_2(y) = 1$. We then further define $g(x,y)$ for convenience, which only contains non-negative powers of $x$ and $y$. Now, consider the fractal pattern generated by

$$\bar{x}^a\bar{y}\left[1 + \bar{g}(x,y)\bar{x} + \bar{g}(x,y)^2\bar{x}^2 + \dots\right], \tag{18}$$

which describes a higher-order CA evolving in the $\bar{x}$ direction. Note that powers of $\bar{g}$ no longer have the nice interpretation of representing an equal time state in terms of this CA, due to it containing both powers of $\bar{y}$ as well as $\bar{x}$ (but evaluating the series up to the $\bar{g}^n\bar{x}^n$ does give the correct configuration up to $\bar{x}^n$). As $\bar{g}$ contains only *negative* powers of $y$, this fractal pattern is restricted only to the half-plane with $y^{j<0}$. It thus lives entirely in the "past", $t < 0$, of our initial CA.

The full fractal given by

$$\mathcal{F}(x,y) = \left[\sum_{l=0}^{\infty}(fy)^l\right] + \bar{x}^a\bar{y}\left[\sum_{l=0}^{\infty}(\bar{g}\bar{x})^l\right] \tag{19}$$

unambiguously describes a history of the CA with the $t = 0$ state $s_0 = 1$. This is shown in Figure 3 for the Fibonacci model, with the forward propagation of $f$ in red and the propagation of $\bar{g}$ in orange.

Going back to operator language, it can be shown straightforwardly that the symmetry action

$$S(q) = X(q(x)\mathcal{F}(x,y)) \tag{20}$$

for arbitrary $q(x)$ commutes with the Hamiltonian (Eq 10 but with all $ij$ included in the sum) everywhere. The only term with $y^0$ in $\mathcal{F}$ is 1, so this operator only flips the spins $q(x)$ on row $y^0$. Furthermore, the choice of choosing the $y^0$ row for defining this symmetry does not

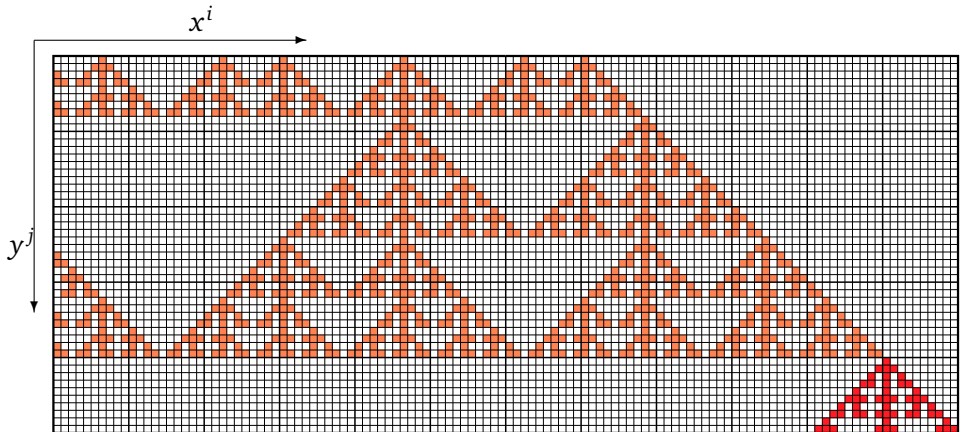

Figure 3: A valid history for the state $s_0 = 1$ for the Fibonacci rule CA. The forward evolution (red) is fully deterministic, and here an unambiguous choice has been made for states leading up to it (orange). Lattice points are labeled by $(i, j)$ corresponding to $x^i y^j$ in the polynomial representation.

affect which operators can be generated, as it is easy to show that $f(x)\mathscr{F}(x, y) = \bar{y}\mathscr{F}(x, y)$, so that $S(q(x)f(x))$ flips any set of spins $q(x)$ on the row $\bar{y}$ instead. Simple counting would then suggest that the total number of symmetry generators thus scales *linearly* with the size of the system, like on the semi-infinite cylinder.

This result seems to contradict the irreversibility of the CA. It would suggest that one can fully determine $s_t$ at time $t < 0$ by choosing the state $s_0$ appropriately, which would seemingly imply that the evolution is always reversible. The resolution to this paradox lies in the fact that we are on an infinite lattice, and in this procedure we have chosen the particular $f^{-1}$ such that it only contains finitely positive powers of $x$ (there are in general multiple inverses $f^{-1}$). Defining $h(x) = [g(x, y)]_{y^0}$ such that $f = x^a(1 + \bar{h}\bar{x})$, then we are choosing the inverse

$$f^{-1}(x) = \bar{x}^a(1 + \bar{h}\bar{x} + (\bar{h}\bar{x})^2 + \dots), \tag{21}$$

from which it can be readily verified that $f^{-1}f = 1$. In this language, $\mathscr{F}(x, y)$ looks like

$$\mathscr{F}(x, y) = \cdots + (f^{-1}\bar{y})^2 + (f^{-1}\bar{y}) + 1 + (fy) + (fy)^2 + \dots, \tag{22}$$

which obviously commutes with the Hamiltonian. As an example, with the Sierpinski rule, the two possible histories for the state $s_0 = 1$ are $s_{-1}^{(-)} = \sum_{l=-1}^{-\infty} x^l$ and $s_{-1}^{(+)} = \sum_{l=0}^{\infty} x^l$. By this inverse, we would only get $s_{-1}^{(-)}$. However, if we wanted to generate the state with history $s_{-1}^{(+)}$, we would instead find that the $t = 0$ state should be the limit $s_0 = 1 + x^\infty$. If we were just interested in any finite portion of the infinite lattice, for example, we may get any history by simply pushing this $x^\infty$ beyond the boundaries.

## 3.5 Open slab

Finally, consider the system on an open slab with dimensions $L_x \times L_y$. Elements of the symmetry group are in correspondence with valid CA configurations on this geometry. The state at time $t = 0$ may be chosen arbitrarily, giving us $L_x$ degrees of freedom. Furthermore, at each time step the state of the cells near the edge may not be fully specified by the CA rules. Hence, each of these adds an additional degree of freedom. Let $x^{-p_{\min}}$, $x^{p_{\max}}$, be the smallest and largest powers of $x$ in $f$ (if $p_{\min/\max}$ would be negative, then set set it to 0). Then, we are free to choose the cell states in a band $p_{\max} \times L_y$ along the left ($x^{i=0}$) edge, and $p_{\min} \times L_y$ along the right edge as well. Thus, the total number of choices is $N_{\text{sym}} = 2^{L_x + (p_{\min} + p_{\max})(L_y - 1)}$,

and there are $\log_2 N_{\mathrm{sym}}$ *independent* symmetry generators. Note that some of these symmetries may be localized to the corners.

One may be tempted to pick a certain boundary condition for the CA, for example, by taking the state of cells outside to be 0, which eliminates the freedom to choose spin states along the edge and reduces the order of the symmetry group down to simply $2^{L_x}$. What will happen in this case is that there will be symmetry elements from the full *infinite* lattice symmetry group which, when restricted to an $L_x \times L_y$ slab, will not look like any of these $2^{L_x}$ symmetries. With the first choice, we are guaranteed that any symmetry of the infinite lattice, restricted to this slab, will look like one of our $N_{\mathrm{sym}}$ symmetries. This is a far more natural definition, and will be important in our future discussion of edge modes in Sec 5.3.

# 4 Spontaneous fractal symmetry breaking

At $T = 0$, the ground state of $\mathcal{H}_{\mathrm{classical}}$ is $2^k$-degenerate and spontaneously breaks the fractal symmetries, where $k$ is the number of independent symmetry generators (which will depend on system size and choice of boundary conditions). Note that $k$ will scale at most linearly with system size, so it represents a subextensive contribution of the thermodynamic entropy at $T = 0$. As a diagnosis for long range order, one has the many-body correlation function $C(\ell)$ given by

$$C(\ell) \;=\; Z\left((1+\bar{f}\,\bar{y})\sum_{i=0}^{\ell-1}(\bar{f}\,\bar{y})^i\right) = Z(1+(\bar{f}\,\bar{y})^\ell), \qquad (23)$$

which has $C(\ell) = 1$ in the ground states of $\mathcal{H}_{\mathrm{classical}}$ as can be seen by the fact that Eq 23 is a product of terms in the Hamiltonian. If $M$ is the number of terms in $f$, then this becomes an $M+1$-body correlation function when $\ell = 2^l$ is a power of 2. Long range order is diagnosed by $\lim_{\ell\to\infty} C(\ell) = \mathrm{const}$. At any finite temperature, however, these models are disordered and have $C(\ell)$ vanishing asymptotically as $C(\ell) \sim p^{-\ell^d}$, where $d$ is the Hausdorff dimension of the generated fractal, and $p = 1/(1+e^{-2\beta})$. This can be seen by mapping to the dual (defect) variables in which the Hamiltonian takes the form of a simple non-interacting paramagnet [70], and the correlation function $C(\ell)$ maps on to a $\mathcal{O}(\ell^d)$-body correlation function. Thus, there is no thermodynamic phase transition in any of these models, although the correlation length defined through $C(\ell)$ diverges as $T \to 0$.

Even without a thermodynamic phase transition, much like in the standard Ising chain, there is the possibility of a quantum phase transition at $T = 0$. We may include quantum fluctuations via the addition of a transverse field $h$,

$$H_{\mathrm{Quantum}} = -\sum_{ij} Z(x^i y^j[1+\bar{f}\,\bar{y}]) - h\sum_{ij} X(x^i y^j). \qquad (24)$$

One can confirm that a small $h$ will indeed correspond to a finite correction $\lim_{l\to\infty} C(2^l) = 1 - \mathrm{const}(h)$, and so does not destroy long range order. This model now exhibits a zero-temperature quantum phase transition at $h = 1$, which is exactly pinpointed by a Kramers-Wannier type self-duality transformation which exchanges the strong and weak-coupling limits. This self-duality is readily apparent by examining the model in terms of defect variables, which interchanges the role of the coupling and field terms. This should be viewed in exact analogy with the 1D Ising chain, which similarly exhibits a $T = 0$ quantum phase transition but fails to have a thermodynamic phase transition.

The transition at $h = 1$ is a spontaneous symmetry breaking transition in which all $2^k$ fractal symmetries are spontaneously broken at once (although under general perturbations they do

not have to all be broken at the same time). Numerical evidence [69] suggests a first order transition. If one were to allow explicitly fractal symmetry breaking terms in the Hamiltonian ($Z$-fields, for example) then it is possible to go between these two phases adiabatically. Thus, as long as the fractal symmetries are not explicitly broken in the Hamiltonian, these two phases are properly distinct in the usual picture of spontaneously broken symmetries. In the following, we will only be discussing ground state ($T = 0$) physics.

## 5 Fractal symmetry protected topological phases

Rather than the trivial paramagnet and spontaneously symmetry broken phases, we may also generate cluster states [99] which are symmetric yet distinct from the trivial paramagnetic phase. These cluster states have the interpretation of being "decorated defect" states, in the spirit of Ref [100], as we will demonstrate. These fractal symmetry protected topological phases (FSPT) are similar to recently introduced subsystem SPTs [63], and were hinted at in Ref 36. In contrast to the subsystem SPTs, however, there is nothing here analogous to a "global" symmetry — the fractal symmetries are the only ones present!

### 5.1 Decorated defect construction

To describe these cluster Hamiltonians, we require a two-site unit cell, which we will refer to as sublattice $a$ and $b$. For the unit cell $(i, j)$ we have two sets of Pauli operators $\hat{Z}_{ij}^{(a)}$, $\hat{Z}_{ij}^{(b)}$, and similarly $\hat{X}_{ij}^{(a/b)}$ and $\hat{Y}_{ij}^{(a/b)}$. Our previous polynomial representation is extended as

$$
Z\begin{pmatrix} \alpha \\ \beta \end{pmatrix} = Z\begin{pmatrix} \sum_{ij} c_{ij}^{(a)} x^i y^j \\ \sum_{ij} c_{ij}^{(b)} x^i y^j \end{pmatrix} = \prod_{ij} \left(\hat{Z}_{ij}^{(a)}\right)^{c_{ij}^{(a)}} \left(\hat{Z}_{ij}^{(b)}\right)^{c_{ij}^{(b)}}, \tag{25}
$$

and similarly for $X(\cdot)$ and $Y(\cdot)$. This notation is easily generalized to $n$ spins per unit cell, represented by $n$ component vectors.

Our cluster FSPT Hamiltonian is then given by

$$
\mathscr{H}_{\text{FSPT}} = -\sum_{ij} Z\begin{pmatrix} x^i y^j (1 + \bar{f}\bar{y}) \\ x^i y^j \end{pmatrix} - \sum_{ij} X\begin{pmatrix} x^i y^j \\ x^i y^j (1 + f y) \end{pmatrix}
$$
$$
- h_x \sum_{ij} X\begin{pmatrix} x^i y^j \\ 0 \end{pmatrix} - h_z \sum_{ij} Z\begin{pmatrix} 0 \\ x^i y^j \end{pmatrix}, \tag{26}
$$

which consists of commuting terms and is exactly solvable at $h = h_x = h_z = 0$, which we will assume for now. There is a unique ground state on a torus (regardless of the symmetries). The ground state is short range entangled, and may be completely disentangled by applications of controlled-X (CX) gates at every bond between two different-sublattice sites that share an interaction, as per the usual cluster states — however, this transformation does not respect the fractal symmetries of this model. These fractal symmetries come in two flavors, one for each sublattice:

$$
\mathbb{Z}_2^{(a)} \quad : \quad S^{(a)}(q(x)) = X\begin{pmatrix} q(x)\mathscr{F}(x, y) \\ 0 \end{pmatrix},
$$
$$
\mathbb{Z}_2^{(b)} \quad : \quad S^{(b)}(q(x)) = Z\begin{pmatrix} 0 \\ q(x)\bar{\mathscr{F}}(x, y) \end{pmatrix}, \tag{27}
$$

where we have assumed an infinite plane with $\mathscr{F}(x, y)$ as in Eq 22, and $q(x)$ may be any polynomial.

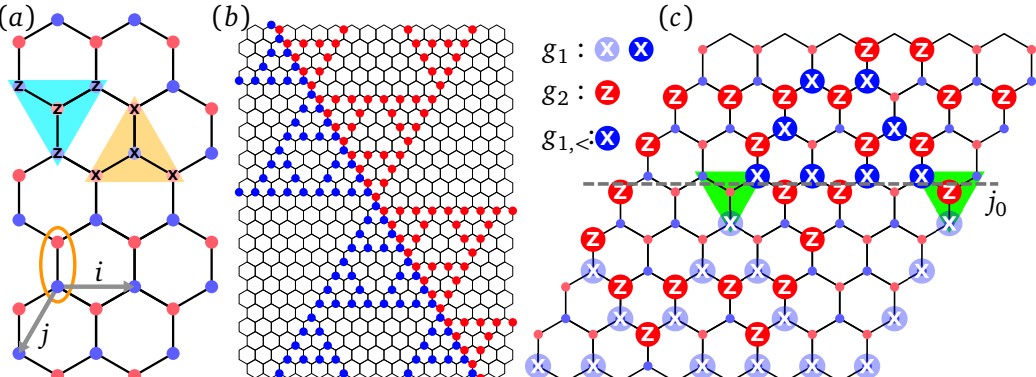

Figure 4: In $(a)$, we show how to place the Sierpinski FSPT on to the honeycomb lattice naturally. The orange circle is the unit cell, and blue/red sites correspond to the $a/b$ sublattice sites. The interactions involve four spins on the highlighted triangles triangles. In $(b)$, we show the sites affected by a choice of symmetry operations on an infinite plane. The large circles are those affected by a particular $\mathbb{Z}_2^{(a/b)}$ type symmetry (Eq 27). In $(c)$, we perform a symmetry twist on the Sierpinski FSPT on a $7 \times 7$ torus. The chosen symmetries $g_1$ $(g_2)$ corresponds to operations on all spins highlighted by a large blue (red) circle. The green triangles correspond to terms in the twisted Hamiltonian $H_{\text{twist}}(g_1)$ that have flipped sign. The charge response $T(g_1, g_2) = -1$ is given by the parity of red circles that also lie in the green triangles, and is independent of where we make the cut $j_0$.

The picture of the ground state is as follows. Working in the $\hat{Z}^{(a)}$, $\hat{Z}^{(b)}$ basis, notice that if $\hat{Z}_{ij}^{(b)} = 1$, the first term in the Hamiltonian simply enforces the $\hat{Z}_{ij}^{(a)}$ spins to follow the standard CA evolution. At locations where $\hat{Z}_{ij}^{(b)} = -1$, there is an "error", or *defect,* of the CA, where the opposite of the CA rule is followed. The second term in the Hamiltonian transitions between states with different configurations of such defects. The ground state is therefore an equal amplitude superposition of all possible configurations. The same picture can also be obtained from the $\hat{X}^{(a)}$, $\hat{X}^{(b)}$ basis, in terms of the CA rules acting on the $\hat{X}_{ij}^{(b)}$ spins.

### 5.1.1 Sierpinski FSPT

As a particularly illustrative example, let us consider the FSPT generated from the Sierpinski rule. The resulting model is the "decorated defect" NM paramagnet, which we refer to as the Sierpinski FSPT. The Hamiltonian is given by

$$\mathcal{H}_{\text{Sier-FSPT}} = -\sum_{ij} \hat{Z}_{ij}^{(a)} \hat{Z}_{i,j-1}^{(a)} \hat{Z}_{i-1,j-1}^{(a)} \hat{Z}_{ij}^{(b)} - \sum_{ij} \hat{X}_{ij}^{(b)} \hat{X}_{i,j+1}^{(b)} \hat{X}_{i+1,j+1}^{(b)} \hat{X}_{ij}^{(a)}. \tag{28}$$

It is particularly enlightening to place this model on a honeycomb lattice, as shown in Fig 4a. Fig 4b shows the action of two symmetries as an example.

We may then redefine $\hat{Z}_{ij}^{(b)} \leftrightarrow \hat{X}_{ij}^{(b)}$, after which the Hamiltonian takes the particularly simple form of a cluster model

$$\mathcal{H}_{\text{cluster}} = -\sum_{s} \hat{X}_s \prod_{s' \in \Gamma(s)} \hat{Z}_{s'}, \tag{29}$$

where $s = (i, j, a/b)$ labels a site on the honeycomb lattice and $\Gamma(s)$ is the set of its nearest neighbors. However, we will generally not use such a representation. Note that this model is isomorphic to the 2D fractal SPT obtained in Ref [107].

### 5.1.2 Fibonacci FSPT

Our other example is the Fibonacci FSPT. The Hamiltonian takes the form

$$
\mathcal{H}_{\text{Fib-FSPT}} = -\sum_{ij} \hat{Z}_{ij}^{(a)} \hat{Z}_{i-1,j-1}^{(a)} \hat{Z}_{i,j-1}^{(a)} \hat{Z}_{i+1,j-1}^{(a)} \hat{Z}_{ij}^{(b)} \tag{30}
$$

$$
- \sum_{ij} \hat{X}_{ij}^{(b)} \hat{X}_{i+1,j+1}^{(b)} \hat{X}_{i,j+1}^{(b)} \hat{X}_{i-1,j+1}^{(b)} \hat{X}_{ij}^{(a)}, \tag{31}
$$

which we illustrate in Fig 5a. Unlike with the Sierpinski FSPT, this model does not have as nice of an interpretation of being a cluster model with interactions among sets of nearest neighbors on some simple 2D lattice.

## 5.2 Symmetry Twisting

To probe the nontriviality of the FSPT symmetric ground state, we may place it on a torus and apply a symmetry twist to the Hamiltonian, and observe the effect in the charge of another symmetry [102–105]. To be concrete, let $\mathcal{H}_{\text{twist}}(g_1)$ be the $g_1$ symmetry twisted Hamiltonian. The $g_2$ charge of the ground state of $\mathcal{H}_{\text{twist}}(g_1)$ relative to its original value tells us about the nontriviality of the phase under these symmetries. That is, let

$$
\langle g_2 \rangle_{g_1} = \lim_{\beta \to \infty} \frac{1}{\mathscr{Z}} \text{Tr}\left[ g_2 e^{-\beta \mathcal{H}_{\text{twist}}(g_1)} \right], \tag{32}
$$

with $\mathscr{Z}$ the partition function, then, we define the charge response

$$
T(g_1, g_2) = \langle g_2 \rangle_{g_1} / \langle g_2 \rangle_{\mathbf{1}}, \tag{33}
$$

where $\langle g_2 \rangle_{\mathbf{1}}$ is simply the $g_2$ charge of the ground state of the untwisted Hamiltonian. On a torus, we may twist along either the horizontal or vertical direction — here we first consider twisting along the vertical direction.

Let us be more concrete. Take the FSPT Hamiltonian (Eq 26) on an $L_x \times L_y$ torus, and let $k$ be the number of independent symmetry generators of the type $\mathbb{Z}_2^{(a)}$ (which is also the same as for $\mathbb{Z}_2^{(b)}$). We assume $L_x, L_y$ have been chosen such that $k > 0$. The total symmetry group of our Hamiltonian is therefore $\left( \mathbb{Z}_2^{(a)} \times \mathbb{Z}_2^{(b)} \right)^k$. Let us label the $2k$ generators for this group

$$
S_\alpha^{(a)} = X \begin{pmatrix} q_\alpha^{(a)}(x) \sum_{l=0}^{L_y-1} (fy)^l \\ 0 \end{pmatrix}; \quad S_\alpha^{(b)} = Z \begin{pmatrix} 0 \\ q_\alpha^{(b)}(x) \sum_{l=0}^{L_y-1} (\bar{f}\bar{y})^l \end{pmatrix}, \tag{34}
$$

where $0 \le \alpha < k$ and $q_\alpha^{(a/b)}(x)$ have been chosen such that the set of $S_\alpha^{(a/b)}$ are all independent. Recall from Section 3.3 that only certain such polynomials $q(x)$ are allowed on a torus.

To apply a $g$-twist, we first express the Hamiltonian as a sum of local terms $\mathcal{H}_{\text{FSPT}} = \sum_{ij} H_{ij}$. We then pick a horizontal cut $j = j_0$, dividing the system between $j < j_0$ and $j \ge j_0$. For each term that crosses the cut, we conjugate $H_{ij} \to g_< H_{ij} g_<^{-1}$, where $g_<$ is the symmetry action of $g$ restricted to $j < j_0$. For an Ising system, this will simply have the effect of flipping the sign of some terms in the Hamiltonian. The resulting Hamiltonian is $\mathcal{H}_{\text{twist}}(g)$.

To understand which terms in the Hamiltonian change sign under conjugation, consider the choice of symmetry $g_1$ in Fig 4c, which consists of flipping all spins in the large blue (dark and transparent) circles. Restricting $g_1$ to $j < j_0$ leaves $g_{1,<}$, flipping only spins in the dark circles. Conjugating by $g_{1,<}$ results in the terms in the green triangles appearing in $\mathcal{H}_{\text{twist}}(g_1)$ with a relative minus sign.

Doing this explicitly for a symmetry element $S_\alpha^{(a)}$, we find that the incomplete symmetry restricted to $j < j_0$ is given by

$$S_{\alpha,<}^{(a)} = X \begin{pmatrix} q_\alpha^{(a)}(x) \sum_{l=0}^{j_0-1}(fy)^l \\ 0 \end{pmatrix}. \tag{35}$$

The terms in the Hamiltonian that pick up a minus sign when conjugated with $S_{\alpha,<}^{(a)}$ are exactly translations of the first term in $\mathscr{H}_{\text{FSPT}}$ (Eq 26) given by the non-zero coefficients of the commutation polynomial along $j_0$: $P = q_\alpha^{(a)}(x)(fy)^{j_0}$. However, the same twisted Hamiltonian may also be obtained by conjugating the entire $\mathscr{H}_{\text{FSPT}}$ by

$$K_\alpha^{(a)} = X \begin{pmatrix} 0 \\ q_\alpha^{(a)}(x)(fy)^{j_0} \end{pmatrix} \tag{36}$$

such that $H_{\text{twist}}(S_\alpha^{(a)}) = K_\alpha^{(a)} \mathscr{H}_{\text{FSPT}} K_\alpha^{(a)\dagger}$.

Next, we can compute the charge of another symmetry $S_\beta^{(a/b)}$ in the ground state of $H_{\text{twist}}(S_\alpha^{(a)})$. Without any twisting, the ground state is uncharged under all symmetries, $\langle S_\alpha^{(a/b)}\rangle_1 = 1$. After the twist, none of $S_\beta^{(a)}$ will have picked up a charge (as they commute with $K_\alpha^{(a)}$), but some $S_\beta^{(b)}$ may pick up a nontrivial charge if they anticommute with $K_\alpha^{(a)}$. Letting $T(S_\alpha^{(a)}, S_\beta^{(b)}) = (-1)^{T_{\alpha\beta}}$, we have

$$T_{\alpha\beta} = \left[ q_\alpha^{(a)}(x)(fy)^{j_0} \times \bar{q}_\beta^{(b)}(x) \sum_{l=0}^{L_y-1}(fy)^l \right]_{x^0 y^0} = \left[ q_\alpha^{(a)}(x)\bar{q}_\beta^{(b)}(x) \right]_{x^0}, \tag{37}$$

where we have used $y^{L_y} = 1$ and the definition of a symmetry on the torus, Eq 15. As expected, the result is independent of our choice of $j_0$, and it is also apparent that $T(g_1, g_2) = T(g_2, g_1)$ for any $g_1, g_2$. If we choose the same symmetry basis for both sublattices, $q_\alpha^{(a)}(x) = \bar{q}_\alpha^{(b)}(x)$, then we additionally get that $T_{\alpha\beta} = T_{\beta\alpha}$.

Figure 4c is an illustration of this twisting calculation for the Sierpinski FSPT on a $7 \times 7$ torus. Letting $x^0 y^0$ label the unit cell in the top left of the figure, $g_1$ is an $(a)$ type symmetry with $q^{(a)}(x) = x^3 + x^4$ and $g_2$ is a $(b)$ type symmetry with $q^{(b)}(x) = x^4 + x^5$. Then, Eq 37 gives $T(g_1, g_2) = -1$, which can be confirmed by eye in the figure.

The exact same procedure may also be applied for twists across the horizontal direction, which will provide yet another set of independent relations between the symmetries (but will not have as nice of a form).

## 5.3 Degenerate edge modes

Upon opening boundaries, the ground state manifold becomes massively degenerate. Away from a corner, we will show that these degeneracies cannot be broken by local perturbations as long as the fractal symmetries are all respected, much like in the case of SPTs with one-dimensional subsystem symmetries [63].

Let us review the open slab geometry from Sec 3.5 for the FSPT. We take the system to be a rectangle with $L_x \times L_y$ unit cells, such that we are restricted to $x^{0 \le i < L_x} y^{0 \le j < L_y}$. as before, let $x^{-p_{\min}}, x^{p_{\max}}$, be the smallest and largest powers of $x$ in $f$ (and let $p_{\min/\max} = 0$ if they would be negative). The total symmetry group is $\left(\mathbb{Z}_2^{(a)} \times \mathbb{Z}_2^{(b)}\right)^k$ with

$$k = L_x + R(L_y - 1); \quad R = p_{\min} + p_{\max}, \tag{38}$$

and we assume $L_x > R$ (otherwise there are no allowed terms in the Hamiltonian at all). A $\mathbb{Z}_2^{(a)}$ type symmetry acts as $\prod \hat{X}_{ij}^{(a)}$ on a subset of unit cells, and a particular symmetry is fully specified by how it acts on the top row $x^i y^0$, the band $x^{i<p_{max}} y^j$ (on the left side), and the band $x^{i \geq L_x - p_{min}} y^j$ (on the right side). A $\mathbb{Z}_2^{(b)}$ type symmetry acts as $\prod \hat{Z}_{ij}^{(b)}$ and a particular one is fully specified in a similar manner, but spatially inverted (top$\leftrightarrow$bottom, left$\leftrightarrow$right). Alternatively, we may simply think of the symmetries as those of the infinite plane, but truncated to the $L_x \times L_y$ slab.

On the open slab, we take our Hamiltonian (Eq 26) with $h = 0$ on the infinite plane and simply exclude terms that contain sites outside of the sample. For each term with shift $x^i y^j$ that are excluded, but for which the unit cell $x^i y^j$ is still in the system, we lose a constraint on the ground state manifold and hence gain a two-fold degeneracy. The number of terms excluded is given by exactly the same counting as before. Along the top (bottom) edge, there is one excluded $Z$ ($X$) term per unit cell. Along the left edge, there are $p_{max}$ $Z$ terms excluded and $p_{min}$ $X$ terms, for a total of $R$ excluded terms per unit cell, and similarly for the right edge. Hence, there are a total of $2^{2k}$ ground states, coming from a $2^R$-fold degeneracy per unit cell along the left/right edges, and 2-fold degeneracy per unit cell along the top/bottom (with some correction for overcounting).

To describe the edge physics on the slab geometry, let us introduce some additional notation. Let us denote the truncation of an arbitrary polynomial $p(x, y)$ to the slab as $[p(x, y)]_{slab}$, where only the terms with $x^i y^j$ with $(i, j) \in$ slab are kept, where

$$\text{slab} = [0, L_x - 1] \times [0, L_y - 1] \tag{39}$$

is the set of sites $(i, j)$ which exist on the $L_x \times L_y$ slab, and $[a, b] = \{a, a+1, \dots b\}$. We may further make the distinction between those on the edge or the bulk of the slab. Let us denote two types of bulks, which we denote by $\text{bulk}_a$ and $\text{bulk}_b$,

$$\text{bulk}_a = [p_{max}, L_x - p_{min} - 1] \times [1, L_y - 1], \tag{40}$$
$$\text{bulk}_b = [p_{min}, L_x - p_{max} - 1] \times [0, L_y - 2], \tag{41}$$

such that the Hamiltonian on the slab is given by

$$\mathscr{H}_{slab} = -\sum_{(i,j) \in \text{bulk}_a} Z\begin{pmatrix} x^i y^j (1 + \bar{f} \bar{y}) \\ x^i y^j \end{pmatrix} - \sum_{(i,j) \in \text{bulk}_b} X\begin{pmatrix} x^i y^j \\ x^i y^j (1 + f y) \end{pmatrix}. \tag{42}$$

Finally, we denote the edge simply as those sites in the slab that are not in the bulk,

$$\text{edge}_a = \text{slab} \setminus \text{bulk}_a, \tag{43}$$
$$\text{edge}_b = \text{slab} \setminus \text{bulk}_b. \tag{44}$$

For each excluded $Z$ term in $\mathscr{H}_{slab}$, i.e. each $(i, j) \in \text{edge}_a$, we may define a set of three Pauli operators,

$$\hat{\mathscr{X}}_{ij}^{(a)} = X\begin{pmatrix} 0 \\ x^i y^j \end{pmatrix}; \quad \hat{\mathscr{Z}}_{ij}^{(a)} = Z\begin{pmatrix} [x^i y^j (1 + \bar{f} \bar{y})]_{slab} \\ x^i y^j \end{pmatrix},$$
$$\hat{\mathscr{Y}}_{ij}^{(a)} = Z\begin{pmatrix} [x^i y^j (1 + \bar{f} \bar{y})]_{slab} \\ 0 \end{pmatrix} Y\begin{pmatrix} 0 \\ x^i y^j \end{pmatrix}, \tag{45}$$

and similarly, for each excluded $X$ term at $(i, j) \in \text{edge}_b$, we may define

$$\hat{\mathscr{X}}_{ij}^{(b)} = X\begin{pmatrix} x^i y^j \\ [x^i y^j (1 + f y)]_{slab} \end{pmatrix}; \quad \hat{\mathscr{Z}}_{ij}^{(b)} = Z\begin{pmatrix} x^i y^j \\ 0 \end{pmatrix},$$
$$\hat{\mathscr{Y}}_{ij}^{(b)} = X\begin{pmatrix} 0 \\ [x^i y^j (1 + f y)]_{slab} \end{pmatrix} Y\begin{pmatrix} x^i y^j \\ 0 \end{pmatrix}, \tag{46}$$

where the $[\cdot]_{\text{slab}}$ truncation ensures that only those sites physically in the slab are involved. We will call such operators "edge" Pauli operators. There are $2k$ such sets of edge Pauli operators, one for each excluded term. It may readily be verified that $\hat{\mathscr{X}}_{ij}^{(a/b)}$, $\hat{\mathscr{Y}}_{ij}^{(a/b)}$, and $\hat{\mathscr{Z}}_{ij}^{(a/b)}$ satisfy the Pauli algebra while being independent of and commuting with every term in the Hamiltonian and each other at different sites. They therefore form a Pauli basis for operators which act purely within the $2^{2k}$ dimensional ground state manifold.

In principle, any local perturbation, projected on to the ground state manifold, will have the form of being some local effective Hamiltonian in terms of these edge Pauli operators, and may break the exact degeneracy. However, we wish to consider only perturbations commuting with all fractal symmetries. To deduce what type of edge Hamiltonian is allowed, we must find out how our many symmetry elements act in terms of these edge operators.

Consider the action of a $\mathbb{Z}_2^{(a)}$ symmetry on the slab,

$$S^{(a)}(q(x)) = X \begin{pmatrix} [q(x)\mathscr{F}(x,y)]_{\text{slab}} \\ 0 \end{pmatrix}, \tag{47}$$

which is written as the truncation of a symmetry on an infinite plane to a slab, as discussed in Sec 3.5. By construction, we have that $\mathscr{F}(x,y)(1+fy) = 0$, so we may also write

$$S^{(a)}(q(x)) = X \begin{pmatrix} [q(x)\mathscr{F}(x,y)]_{\text{slab}} \\ [q(x)\mathscr{F}(x,y)(1+fy)]_{\text{slab}} \end{pmatrix}. \tag{48}$$

Let us denote for convenience $\gamma \equiv q(x)F(x,y)$, which can be decomposed into three parts: $\gamma = [\gamma]_{\text{bulk}_b} + [\gamma]_{\text{edge}_b} + [\gamma]_{\text{slab}^c}$, the $\text{bulk}_b$ part, the $\text{edge}_b$ part, and the parts external to the slab (denoted by the complement $\text{slab}^c$). Then, we may be decompose the symmetry action as

$$\begin{aligned} S^{(a)}(q(x)) &= X \begin{pmatrix} [[\gamma]_{\text{bulk}_b} + [\gamma]_{\text{edge}_b} + [\gamma]_{\text{slab}^c}]_{\text{slab}} \\ [([\gamma]_{\text{bulk}_b} + [\gamma]_{\text{edge}_b} + [\gamma]_{\text{slab}^c})(1+fy)]_{\text{slab}} \end{pmatrix} \\ &= X \begin{pmatrix} [\gamma]_{\text{bulk}_b} \\ [[\gamma]_{\text{bulk}_b}(1+fy)]_{\text{slab}} \end{pmatrix} X \begin{pmatrix} [\gamma]_{\text{edge}_b} \\ [[\gamma]_{\text{edge}_b}(1+fy)]_{\text{slab}} \end{pmatrix} X \begin{pmatrix} 0 \\ [[\gamma]_{\text{slab}^c}(1+fy)]_{\text{slab}} \end{pmatrix}. \end{aligned} \tag{49}$$

The first factor, the bulk action, is made out of products of terms in $\mathscr{H}_{\text{slab}}$ (i.e. is an element of the stabilizer group) and therefore acts trivially on the ground state manifold. The second factor acts only on $\text{edge}_b$, and operates within the ground state manifold as a product of $\hat{\mathscr{X}}_{ij}^{(b)}$ edge Pauli operators. The third factor acts only on $\text{edge}_a$ and operates within the ground state manifold as a product of $\hat{\mathscr{X}}_{ij}^{(a)}$ edge Paulis (as $[[\gamma]_{\text{slab}^c}(1+fy)]_{\text{slab}}$ can only have non-zero coefficients with $(i,j) \in \text{edge}_a$). It is somewhat undesirable to have reference to $[\gamma]_{\text{slab}^c}$ (which exists outside of the slab), thus we may use the fact that $0 = \gamma(1+fy)$ and $\gamma = [\gamma]_{\text{slab}} + [\gamma]_{\text{slab}^c}$ to obtain $[\gamma]_{\text{slab}^c}(1+fy) = [\gamma]_{\text{slab}}(1+fy)$. We therefore have that

$$S^{(a)}(q(x)) = (\text{bulk stabilizer}) \times \prod_{(i,j)\in\text{edge}_b} \left[\hat{\mathscr{X}}_{ij}^{(b)}\right]^{[\gamma]_{x^i y^j}} \times \prod_{(i,j)\in\text{edge}_a} \left[\hat{\mathscr{X}}_{ij}^{(a)}\right]^{[[\gamma]_{\text{slab}}(1+fy)]_{x^i y^j}}. \tag{50}$$

In a similar fashion, we may show that a $\mathbb{Z}_2^{(b)}$ type symmetry acts as a product of $\hat{\mathscr{Z}}_{ij}^{(a)}$ edge Paulis in $\text{edge}_a$ and $\hat{\mathscr{Z}}_{ij}^{(b)}$ edge Paulis in $\text{edge}_b$. The action of a specific element in the symmetry group is specified by $[\gamma]_{\text{slab}}$.

We claim that it is always possible to find a particular symmetry element that acts *locally* on one edge in any way (but it will generally extend non-trivially into the bulk and act in complicated way on the other boundaries). For example, for any $(i_0, j_0)$ on the left edge,

there exists a $\mathbb{Z}_2^{(a)}$ symmetry which acts *only* as $\hat{\mathcal{X}}_{i_0 j_0}^{(b)}$ on the left edge, and there is also a $\mathbb{Z}_2^{(b)}$ symmetry which acts *only* as $\hat{\mathcal{Z}}_{i_0 j_0}^{(a)}$ on the left edge (although their action on the other edges may be complicated). There is no non-trivial operator acting on a single edge that commutes with both $\hat{\mathcal{X}}$ and $\hat{\mathcal{Z}}$, and therefore we are prohibited from adding *anything* non-trivial to the effective Hamiltonian on this edge which therefore guarantees that no degeneracy can be broken while respecting all fractal symmetries. Note that we don't even have the possibility of spontaneous symmetry breaking at the surface — even simple $\hat{\mathcal{Z}}\hat{\mathcal{Z}}$ couplings along the edge violate the symmetries. The only way the ground state degeneracy may be broken without breaking the symmetry is by terms which couple edge Paulis along different edges; these terms are either non-local, or located at a corner of the system.

Suppose we have found a particular $\mathbb{Z}_2^{(a)}$ symmetry element $g_1$ and a $\mathbb{Z}_2^{(b)}$ symmetry element $g_2$ which, on the left edge, acts as $\hat{\mathcal{X}}_{i_0 j_0}^{(a)}$ and $\hat{\mathcal{Z}}_{i_0 j_0}^{(a)}$ respectively on the same site $(i_0, j_0)$, and trivially everywhere else on the left edge (but will act non-trivially on the other edges). These are said to form a *projective representation* of $\mathbb{Z}_2^{(a)} \times \mathbb{Z}_2^{(b)}$ on that edge. That is, a linear (non-projective) representation of $\mathbb{Z}_2^{(a)} \times \mathbb{Z}_2^{(b)}$ with generators $g_1$, $g_2$, would have $(g_1 g_2)^2 = 1$. However, if we look at the action on this particular edge, then we have that $(g_1^{\text{edge}} g_2^{\text{edge}})^2 = (\hat{\mathcal{X}}\hat{\mathcal{Z}})^2 = -1$. Since we know that as a whole $g_1$ and $g_2$ must commute, the action of $g_1$ and $g_2$ on the other edges must again anticommute (to cancel out the $-1$ from this edge). Small manipulations of the edges (such as adding or removing sites) or local unitary transformations respecting the symmetry cannot change the fact that the actions of $g_1$ and $g_2$ are realized projectively on this edge.

Near particular corners, some symmetry elements may act essentially locally. As a symmetry element (as a whole) must commute with all others, nothing prevents the addition of the full symmetry action *itself* as a term in the effective Hamiltonian when it is local. For example, when $h \neq 0$ there will be terms appearing in the effective Hamiltonian at finite order in perturbation theory near such corners, which commute with all symmetries. The magnitude of such terms will decay exponentially away from a corner, however, and therefore we still have an effective degeneracy per unit length along the boundaries.

### 5.3.1 Local action of symmetries on edges

To prove our claim that there is always a symmetry element which acts locally along an edge, let us first consider finding a particular $\mathbb{Z}_2^{(a)}$ symmetry element which acts locally on an edge as $\hat{\mathcal{X}}_{i_0 j_0}^{(a/b)}$. The ability to find a $\mathbb{Z}_2^{(b)}$ symmetry element acting locally as well then follows. Such a symmetry will act locally in some way on the edge, but extend into the bulk in a non-trivial way. Note that there is no "most natural basis" for these symmetries, unlike in the case of integer $d$ subsystem symmetries [63].

Let us take a general $\mathbb{Z}_2^{(a)}$ symmetry element defined according to Eq 47 in terms of a single polynomial $q(x)$. However, multiple $q(x)$ may lead to the same symmetry element on the slab. Recall that in the CA picture, $q(x)$ corresponds to the CA state at time 0, and the $L_x \times L_y$ slab is a space-time trajectory for the CA. There is a strictly defined "light-cone" determined by the CA rules for which cells at time 0 can affect a future cell in our $L_x \times L_y$ slab. It is easy to verify that only the coefficients in $q(x)$ of $x^i$ for $-p_{\max}(L_y - 1) \leq i < L_x + p_{\min}(L_y - 1)$ can affect the way the symmetry acts within the slab. Let us therefore take $q(x)$ to only contain powers of $x$ within this range. Furthermore, we see that there are $2^{p_{\max}(L_y - 1) + L_x + p_{\min}(L_y - 1)} = 2^{L_x + R(L_y - 1)} = 2^k$ possible $q(x)$s, which is also the order of the $(\mathbb{Z}_2^{(a)})^k$ symmetry group, which means that there is a one-to-one correspondence between $q(x)$s and different elements of the symmetry group.

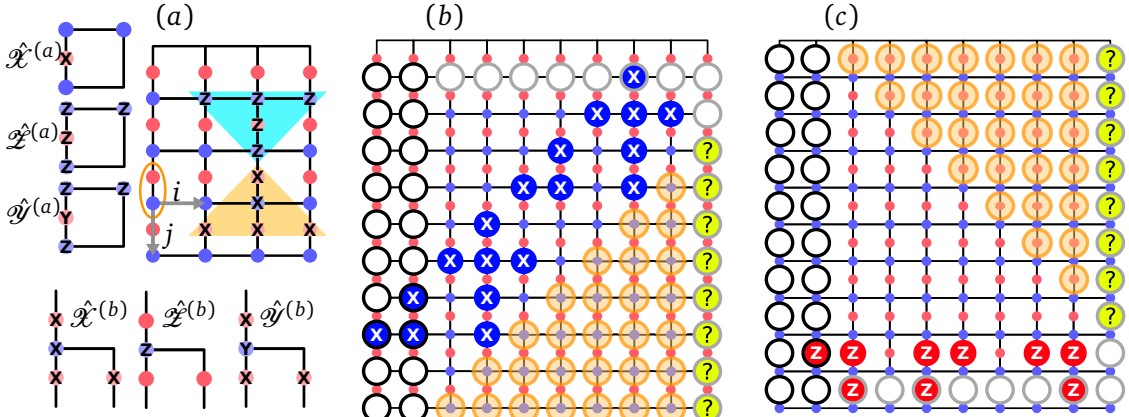

Figure 5: (*a*) We illustrate the terms in the Hamiltonian for the Fibonacci FSPT (Eq 26 with $f = x^{-1} + 1 + x$). The model is defined on a square lattice, with a two-site unit cell (circled), $a$ (blue) and $b$ (red). The two terms in the Hamiltonian at $h = 0$ are illustrated in the two triangles. Also shown are the edge Pauli operators along the left edge. (*b*) We show a family of symmetry elements on a $10 \times 10$ slab. The black outlined circles represent the band of $R = 2$ unit cells on which we fix the action of the symmetry so that it acts only as $\hat{\mathcal{X}}_{0,7}^{(b)}$ on the left edge in this case (with $(0,0)$ being the top left unit cell). This fixes how the symmetry must act on the top and some of the right edge (gray outlined circles), but there is still some freedom along the remaining sites on the right edge (yellow question marks), which will determine how it acts on the remaining sites (transparent orange circles). There are $2^{L_x - R} = 2^8$ symmetry elements (corresponding to the 8 question marks) satisfying our constraint. (*c*) We also show the family of symmetry elements which act only as $\hat{\mathcal{Z}}_{0,7}^{(b)}$, and therefore forms a projective representation with the symmetry element shown in (*b*) on the left edge. Note that these symmetries will generally have some non-trivial action along the other edges.

**Top edge**    Finding a symmetry element that acts locally on the top edge is simple. The only possibilities on the top edge are for it to act as $\hat{\mathcal{X}}_{i,0}^{(a)}$ operators. For example, we may simply choose any $q(x)$ such that $[q(x)]_{\text{slab}} = x^{i_0}$, and the corresponding symmetry element will act locally as only $\hat{\mathcal{X}}_{i_0,0}^{(a)}$ on the top edge (here $[\cdot]_{\text{slab}}$ simply means we keep only the terms with $x^{0 \le i < L_x}$).

**Bottom edge**    Along the bottom edge, the only possibility is for a symmetry elemnt to act as $\hat{\mathcal{X}}_{i,L_y-1}^{(b)}$. Any $q(x)$ chosen such that $[q(x)f^L]_{\text{slab}} = x^{i_0}$ will act locally as only $\hat{\mathcal{X}}_{i_0,L_y-1}^{(b)}$ on the bottom edge. There is always such a $q(x)$ that does this, as we showed for the infinite plane (Sec 3.4) that one can always find a history for any CA state.

**Left/right edge**    Along the left/right edges, things are slightly trickier. Let us look at only the left edge for now. A symmetry element may act as $\hat{\mathcal{X}}_{ij}^{(a)}$ for $0 \le j < p_{\max}$, or as $\hat{\mathcal{X}}_{ij}^{(b)}$ for $0 \le j < p_{\min}$. Per unit cell along the left edge, there are $2^{p_{\min}+p_{\max}} = 2^R$ possible ways to act. From Eq 50, we see that the non-zero coefficients of $q(x)\mathcal{F}(x,y)$ in the columns $0 \le j < p_{\min}$ of the slab directly correspond to how the symmetry element acts as $\hat{\mathcal{X}}_{ij}^{(a)}$ on the left edge. Once these have been fixed, the coefficients on the $p_{\min} \le j < R$ columns must be chosen to specify how the symmetry element acts (as $\hat{\mathcal{X}}_{ij}^{(b)}$) on the left edge. Thus, to find a particular symmetry element that will act in a particular way on the left edge, we must specify the leftmost $R$ columns of $q(x)\mathcal{F}(x,y)$. By a similar lightcone argument as before, these $R$ columns are

affected by coefficients of $x^i$ in $q(x)$ with $-p_{\max}(L_y-1) \leq i < R + p_{\min}(L_y-1)$. As there are a total of $2^{RL_y}$ possible histories, and also $2^{RL_y}$ cells within the leftmost $R$ columns, we may fully specify the action of the symmetry within these leftmost $R$ columns by an appropriate choice of $q(x)$. The remaining degrees of freedom in $q(x)$ means that there are a total of $2^{k-RL_y} = 2^{L_x-R}$ symmetry elements that act in the same way on the left edge.

Figure 5(right) shows the family of $\mathbb{Z}_2^{(a)}$ symmetry elements chosen to act as only one $\hat{\mathscr{X}}_{ij}^{(b)}$ on the left edge, for the Fibonacci FSPT (Eq 31), whose terms are shown in Fig 5(left). The freedom to choose how the symmetry acts on the right edge (question marks) exactly corresponds to the $2^{L_x-R}$ symmetry elements with the specified action on the left edge. These form a $\mathbb{Z}_2^{L_x-R}$ subgroup of the total symmetry group. We choose to show the Fibonacci FSPT here rather than the Sierpinski FSPT, as the latter has $R = 1$ and is straightforward.

## 5.4 Excitations

On the infinite plane, the lowest lying excitations are strictly immobile. They are therefore *fractons* protected by the total fractal symmetry group.

Take $h = 0$, the lowest lying excited states consist of excitations of a single term in the Hamiltonian, say the $Z$ term at site $x^0 y^0$. This excited state can be obtained by acting on the ground state with $\hat{X}_{0,0}^{(b)}$. One may alternatively think in terms of symmetries. Take an independent set of symmetry generators $g_\alpha^{(a/b)}$ of the form Eq 27 with the basis choice $q_\alpha^{(a/b)} = x^\alpha$. We find that this excited state is uncharged, $\langle g_\alpha^{(a/b)} \rangle = 1$, with respect to all symmetry elements *except* $g_0^{(b)}$, for which it has $-1$ charge. In fact, the *only* state with a single excitation with $\langle g_0^{(b)} \rangle = -1$ is this one with the excitation at the origin.

Let us consider the block of the Hamiltonian with symmetry charges $\langle g_\alpha^{(b)} \rangle = (-1)^{d_\alpha}$. The blocks containing states with single fractons will have

$$\sum_{\alpha=-\infty}^{\infty} d_\alpha x^\alpha = x^i \bar{f}^j, \tag{51}$$

for which the excitation is strictly localized at site $x^i y^j$. The excitation may move away from $x^i y^j$, but at the cost of creating additional excitations as well, such that the charge of all symmetries are unchanged. If one allows breaking of the fractal symmetries, then these charges are no longer conserved and nothing prevents the excitation from moving to a different site.

On lattices with different topology, these fractons may not be strictly immobile. For example, on a torus, depending on the symmetries, a fracton may be able to move to some subset of other sites (or all other sites, if the symmetry group is trivial). However, such hopping terms are exponentially suppressed with system size. In fact, for the Sierpinski FSPT on a torus with no symmetries, it is actually easier perturbatively to hop a fracton a large power of 2 away than it is to hop a short distance (mimicking some form of $p$-adic geometry with $p = 2$).

On an open slab, the ground state manifold is degenerate and all charge assignments are possible in the ground state, protected by the symmetry. Therefore, a fracton may be created, or moved, in any way. However, the amplitude for doing so will decay exponentially away from the edges, and certain processes may only be possible near certain types of edges or corners. The possibilities will depend on the details of the model.

## 5.5 Duality

Here we outline a duality that exist generally for these models, which maps the FSPT phase to two copies of the spontaneous symmetry broken phase of the quantum Hamiltonian in Sec 4. This duality involves non-local transformations and maps the $2^{2k}$ ground states of the FSPT on the open slab to the $2^{2k}$ symmetry breaking ground states of the dual model.

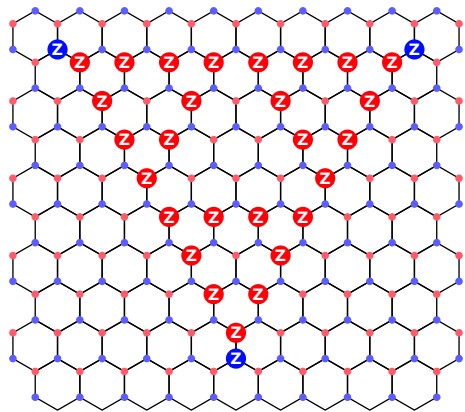

Figure 6: Illustration of the fractal order parameter $C_{\text{FSPT}}(\ell)$ for detecting the FSPT phase of the Sierpinski FSPT, for $\ell = 2^3$. The operator is a product of $Z$ on the highlighted sites.

This duality is most naturally described on an $L_x \times L_y$ cylinder (with $x^{L_x} = 1$) or slab. Let us define new Pauli operators $\tilde{Z}(\cdot)$ and $\tilde{X}(\cdot)$ as

$$
\tilde{Z}\begin{pmatrix} 0 \\ 1 \end{pmatrix} = Z\begin{pmatrix} 0 \\ 1 \end{pmatrix}; \quad \tilde{X}\begin{pmatrix} 1 \\ 0 \end{pmatrix} = X\begin{pmatrix} 1 \\ 0 \end{pmatrix}, \tag{52}
$$

$$
\tilde{Z}\begin{pmatrix} 1 \\ 0 \end{pmatrix} = Z\begin{pmatrix} 1 \\ 1 + \bar{f}\bar{y} + (\bar{f}\bar{y})^2 + \dots \end{pmatrix},
$$

$$
\tilde{X}\begin{pmatrix} 0 \\ 1 \end{pmatrix} = X\begin{pmatrix} 1 + fy + (fy)^2 + \dots \\ 1 \end{pmatrix}, \tag{53}
$$

and translations thereof. It can be readily verified that the latter two commute, and as a whole the set of these operators satisfy the correct Pauli algebra. The fractal symmetries only involve operators in line 52, and so are unchanged. The interaction terms are modified however: in terms of these operators, we have

$$
\tilde{Z}\begin{pmatrix} 1 + \bar{f}\bar{y} \\ 0 \end{pmatrix} = Z\begin{pmatrix} 1 + \bar{f}\bar{y} \\ 1 \end{pmatrix}; \quad \tilde{X}\begin{pmatrix} 0 \\ 1 + fy \end{pmatrix} = X\begin{pmatrix} 1 \\ 1 + fy \end{pmatrix}, \tag{54}
$$

and so the Hamiltonian $\mathcal{H}_{\text{FSPT}}$ (Eq 26) becomes two decoupled copies of $\mathcal{H}_{\text{Quantum}}$ (Eq 24) with their own set of symmetries.

From this, it follows that the order parameter measuring long-range order in $\mathcal{H}_{\text{Quantum}}$, $C(\ell)$ (Eq 23), maps on to a *fractal* order parameter in our original basis

$$
C_{\text{FSPT}}(\ell) = \tilde{Z}\begin{pmatrix} 1 + (\bar{f}\bar{y})^\ell \\ 0 \end{pmatrix} = Z\begin{pmatrix} 1 + (\bar{f}\bar{y})^\ell \\ 1 + \bar{f}\bar{y} + \dots + (\bar{f}\bar{y})^{\ell-1} \end{pmatrix}, \tag{55}
$$

which is pictorially shown for the Sierpinski FSPT in Figure 6, and approaches a constant in the FSPT phase, or zero in the trivial paramagnet, as $\ell = 2^l \to \infty$. By the self-duality of $\mathcal{H}_{\text{Quantum}}$, we also know the FSPT to trivial transition happens at exactly $h = 1$.

Finally, this duality allows us to determine the full phase diagram even as $h_x \neq h_z$. Keeping $h_x$ small and making $h_z$ large, one of the $\mathcal{H}_{\text{Quantum}}$ is driven into its paramagnetic phase where spins are polarized as $\hat{Z}_{ij}^{(b)} = 1$. The Hamiltonian $\mathcal{H}_{\text{FSPT}}$ then looks like a single $\mathcal{H}_{\text{Quantum}}$, and therefore has spontaneously symmetry broken ground states. By the duality transformation, we know this transition happens at exactly $h_z = 1$. The phase diagram is summarized in Fig 7(left).

# 6 Three dimensions

Here, we briefly examine the possible physics available in higher dimension. We consider our symmetry-defining CA in 3D in two ways: via one 2D CA, or two 1D CA. The first will have similar properties to our earlier models, while the latter in certain limits also lead to exotic fractal spin liquids introduced by Yoshida [32] and Haah [33], and may be thought of as (Type-II [37]) symmetry-enriched fracton topologically ordered (FSET) phases.

## 6.1 One 2D cellular automaton

A 2D CA has a two-dimensional state space, combined with one time direction. The state of such a CA may be straightforwardly represented by a polynomial in two variables, $s_t(x,z)$, where the state of the $(i,k)$th cell is given by the coefficient of $x^i z^k$. The update rule is given as a two variable polynomial $f(x,z)$, such that $s_{t+1} = f s_t$ as before. Two dimensional CA also result in a rich variety of fractal structures [101]. The classical Hamiltonian takes the form

$$\mathcal{H}_{1CA} = -\sum_{ijk} Z(x^i y^j z^k [1 + \bar{f}(x,z)\bar{y}]), \tag{56}$$

with symmetries on the semi-infinite system (with $y^{j \geq 0}$) given by

$$S(q(x,z)) = X(q(x,z)[1 + f y + (f y)^2 + \dots]), \tag{57}$$

which commutes with $\mathcal{H}_{1CA}$ everywhere. On an infinite system, an inverse evolution $f^{-1}$ may be defined analogous to Eq 21 and the symmetry action takes the form

$$S(q(x,z)) = X(q(x,z)\mathcal{F}(x,y,z)), \tag{58}$$

with

$$\mathcal{F}(x,y,z) = \dots + (f^{-1}(x,z)\bar{y})^{-2} + f^{-1}(x,z)\bar{y} + 1 + f(x,z)y + (f(x,z)y)^2 + \dots. \tag{59}$$

The discussion of Sec 4 and 5 may then be generalized in a straightforward manner. The phase diagram is exactly the same as in 2D, given by Fig 7(left).

As an example model, consider the Sierpinski Tetrahedron model, given by the update rule $f(x,z) = 1 + x + z$. The Hamiltonian is given by

$$\mathcal{H}_{\text{Sier-Tet}} = -\sum_{ijk} Z_{i,j,k} Z_{i,j-1,k} Z_{i-1,j-1,k} Z_{i,j-1,k-1}. \tag{60}$$

The fractal structure of the symmetries for this model are Sierpinski Tetrahedra, with Hausdorff dimension $d = 2$. The quantum model may be constructed which exhibit the same properties: self-duality about $h = 1$, spontaneous fractal symmetry breaking, and instability to non-zero temperatures. A cluster FSPT version may also be constructed, with the Hamiltonian

$$\begin{aligned} \mathcal{H}_{\text{Sier-Tet-FSPT}} &= -\sum_{ijk} Z^{(a)}_{i,j,k} Z^{(a)}_{i,j-1,k} Z^{(a)}_{i-1,j-1,k} Z^{(a)}_{i,j-1,k-1} Z^{(b)}_{i,j,k} \\ &\quad -\sum_{ijk} X^{(b)}_{i,j,k} X^{(b)}_{i,j+1,k} X^{(b)}_{i+1,j+1,k} X^{(b)}_{i,j+1,k+1} X^{(a)}_{i,j,k}. \end{aligned} \tag{61}$$

This cluster FSPT also has the nice interpretation of being the cluster model (Eq 29) on the diamond lattice. In the presence of an edge, terms in the Hamiltonian must be excluded leading to degeneracies, and in exactly the same way as in 2D one finds these degeneracies along a surface cannot be gapped, thus leading to a $2^{\mathcal{O}(L^2)}$ overall symmetry protected degeneracy for an open system.

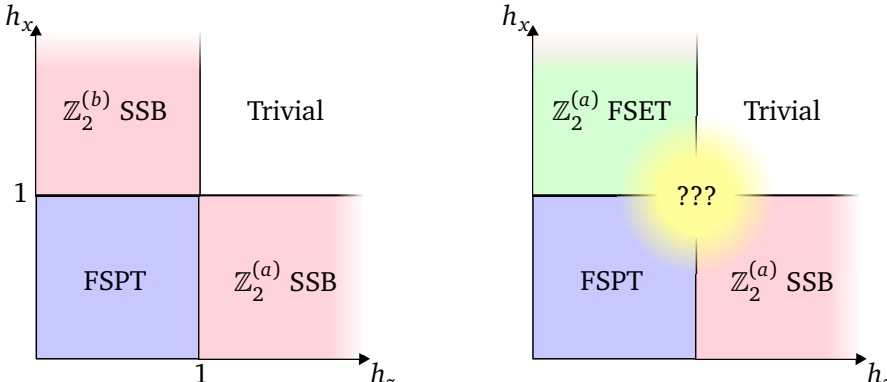

Figure 7: (left) Phase diagram of our 2D or 3D FSPT models generated by one CA, under $h_{x/z} \geq 0$ perturbations. Possible phases include the FSPT phase symmetric under all $\mathbb{Z}_2^{(a)}$ and $\mathbb{Z}_2^{(b)}$ symmetries, two spontaneous symmetry broken (SSB) phases where either of the two types of symmetries are spontaneously broken, and the trivial paramagnetic phase. (right) Sketch of the phase diagram for the 3D models with symmetries generated by two 1D CA. There exists the FSPT phase at small $h_{x/z}$, a SSB phase at large $h_z$, a fracton topologically ordered phase *enriched* with with $\mathbb{Z}_2^{(a)}$ symmetry (FSET) at large $h_x$, and a trivial phase at both large $h_x$ and $h_z$. For this model, we do not know what the phase diagram looks like outside of these limits.

## 6.2  Two 1D cellular automata

Symmetries defined through two 1D CA allow for a wide variety of possibilities. This may be thought of as evolving a 1D CA through two time directions, with potentially different update rules along the two time directions. Let the state of the 1D CA at time $(t_1, t_2)$ be represented by a polynomial $s_{t_1 t_2}(x)$. The update rules along the two time directions are given as two polynomials $f_1(x)$ and $f_2(x)$, with $s_{t_1+1, t_2} = f_1(x)s_{t_1, t_2}$ and $s_{t_1, t_2+1} = f_2(x)s_{t_1, t_2}$. Interpreting the $y, z$, directions as the $t_1, t_2$, directions, the classical 3D Hamiltonian takes the form

$$
\begin{aligned}
\mathcal{H}_{2\text{CA}} &= -\sum_{ijk} Z(x^i y^j z^k \bar{\alpha}) - \sum_{ijk} Z(x^i y^j z^k \bar{\beta}) \\
&= -\sum_{ijk} Z(\bar{\alpha}) - \sum_{ijk} Z(\bar{\beta}),
\end{aligned}
\tag{62}
$$

where $\alpha = 1 + f_1 y$ and $\beta = 1 + f_2 z$ are defined, and in the second line for notational convenience we have suppressed the $x^i y^j z^k$ factor, when summation over translations is apparent (and we will continue to do so). The fractal symmetries on a semi-infinite system (with $x^i y^{j \geq 0} z^{k \geq 0}$ are of the form)

$$
S(q(x)) = X \left( q(x)[1 + f_1 y + (f_1 y)^2 + \dots][1 + f_2 z + (f_2 z)^2 + \dots] \right),
\tag{63}
$$

which can be readily verified to commute with everything in the Hamiltonian. On an infinite system some inverse may again be defined and the symmetry takes the form

$$
S(q(x)) = X(q(x)\mathscr{F}_1(x, y)\mathscr{F}_2(x, z)),
\tag{64}
$$

with $\mathscr{F}_{1/2}$ each defined as in Eq 22 with $f_{1/2}$.

The decorated defect construction starting from $\mathcal{H}_{2\text{CA}}$ results in the following Hamiltonian,



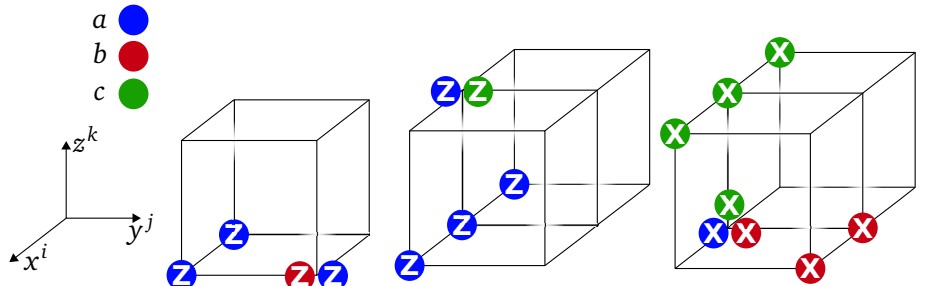

Figure 8: The first three terms in the 3D FSPT Hamiltonian $\mathcal{H}_{\text{FSPT}}$ (Eq 65) generated from two CA, using $f_1 = 1 + x$ the Sierpinski rule and $f_2 = \bar{x} + 1 + x$ the Fibonacci rule. There are three spins on each site of the cubic lattice, labeled by $a$ (blue), $b$ (red), and $c$ (green). Terms are composed of products of $X$ and $Z$ Pauli operators as shown. The Hamiltonian is a sum of translations of these terms.

with three spins per unit cell, on which we have operators $\hat{Z}_{ij}^{(a/b/c)}$ and $\hat{X}_{ij}^{(a/b/c)}$,

$$
\mathcal{H}_{\text{FSPT}} = -\sum_{ijk} Z \begin{pmatrix} \bar{\alpha} \\ 1 \\ 0 \end{pmatrix} - \sum_{ijk} Z \begin{pmatrix} \bar{\beta} \\ 0 \\ 1 \end{pmatrix} - \sum_{ijk} X \begin{pmatrix} 1 \\ \alpha \\ \beta \end{pmatrix}
$$
$$
- \sum_{ijk} \left[ h_x X \begin{pmatrix} 1 \\ 0 \\ 0 \end{pmatrix} + h_z Z \begin{pmatrix} 0 \\ 1 \\ 0 \end{pmatrix} + h_z Z \begin{pmatrix} 0 \\ 0 \\ 1 \end{pmatrix} \right], \tag{65}
$$

which is illustrated in Fig 8, for $f_1 = 1 + x$ and $f_2 = \bar{x} + 1 + x$ (the Sierpinski-Fibonacci model). The first three terms all mutually commute, and $h_x, h_z$ are small perturbations. The symmetries come in three types: first, we still have the original symmetry elements

$$
\mathbb{Z}_2^{(a)} \quad : \quad S^{(a)}(q(x)) = X \begin{pmatrix} q(x)\mathscr{F}_1(x,y)\mathscr{F}_2(x,z) \\ 0 \\ 0 \end{pmatrix},
$$
$$
\tag{66}
$$

but now the remaining independent symmetry elements are more complicated, which arises because there is a further local operator that commutes with the first three terms in $\mathcal{H}_{\text{FSPT}}$, given by

$$
\hat{B}_{ijk} = Z \left( x^i y^j z^k \begin{pmatrix} 0 \\ \bar{\beta} \\ \bar{\alpha} \end{pmatrix} \right). \tag{67}
$$

Due to the existence of $\hat{B}_{ijk}$, given any symmetry operation $S$, $\hat{B}_{ijk}S$ is also a valid symmetry. Thus, these should be thought of as *higher form* fractal symmetries [106]. Consider the analogy with, say, a 1-form symmetries in 3D: these are symmetries which act along a 2 dimensional manifold which may be deformed by local operations. Here, we have the symmetry operations acting on only $b$ or only $c$ sublattice sites which may be made to live on a single plane,

$$
\mathbb{Z}_2^{(b)} \quad : \quad S^{(b)}(q(x,z)) = Z \begin{pmatrix} 0 \\ q(x,z)\bar{\mathscr{F}}_1(x,y) \\ 0 \end{pmatrix},
$$
$$
\mathbb{Z}_2^{(c)} \quad : \quad S^{(c)}(q(x,y)) = Z \begin{pmatrix} 0 \\ 0 \\ q(x,y)\bar{\mathscr{F}}_2(x,z) \end{pmatrix}, \tag{68}
$$

but we are also free to deform such symmetries using products of $\hat{B}_{ijk}$. Such higher form fractal symmetries are an interesting subject by themselves, and we leave a more thorough investigation as a topic for future study.

One may confirm that when $h_x = h_z = 0$, all these symmetries are products of terms in the Hamiltonian, and therefore must have expectation value 1 in the ground state. As every term is independent, and there are three terms that must be satisfied per unit cell of three sites, the ground state is unique. This model in fact describes an FSPT protected by the combination of the "global" fractal symmetries $\mathbb{Z}_2^{(a)}$, along with the set of higher form fractal symmetries $\mathbb{Z}_2^{(b/c)}$. To see this, one may examine the boundary theory. Let's consider the simplest case of $f_1 = f_2 = 1 + x$ the double Sierpinski. On the top surface, with edge Pauli operators $\mathscr{Z}, \mathscr{X}$, one finds that $\mathbb{Z}_2^{(a)}$ acts as a 2D Sierpinski fractal symmetry $S^{(a)} = \prod \mathscr{X}$, while the $\mathbb{Z}_2^{(b/c)}$ symmetries may be chosen to act as $\mathscr{Z}$ on a single site. Thus, the only Hamiltonian we can write down on the surface must be composed of $\mathscr{Z}$ (to commute with a local $\mathscr{Z}$) and must commute with the fractal symmetry. The only possibility is therefore the classical Hamiltonian (as in Eq 10), which exhibits spontaneous fractal symmetry breaking in the ground state. Thus, the surface is non-trivial and must either be gapless or spontaneous symmetry breaking.

Figure 7(right) shows a sketch the phase diagram for this model. Increasing $h_{x/z}$ drives this model out of the FSPT phase. If we increase only $h_z$ while keeping $h_x$ small, we arrive at the spontaneously fractal symmetry broken phase like in the 2D FSPT. Increasing both $h_x$ and $h_z$ too large will result in the trivial paramagnetic phase. However, if we only increase $h_x$ while keeping $h_z$ small, the system enters into a symmetric fracton *topologically ordered* phase, which is the subject of the following discussion.

### 6.2.1 Connection to fracton topological order

The decorated defect approach of the previous sections may be thought of alternatively as the following process:

1. Start with a classical Hamiltonian and some symmetries involving flipping some spins

2. Introduce additional degrees of freedom at each site and couple them to the interaction terms via a cluster-like interaction (this is exactly what one would get following the gauging procedure of Refs [36, 37], and adding the gauge constraint as a term in the Hamiltonian).

3. The resulting theory still has the original symmetries, along with some additional symmetry which we may define acting on the new spins, which we take to be the defining symmetries our model.

4. Perturbations respecting these symmetries may then be added to the Hamiltonian (note these may break the gauge constraint from earlier: we are now interpreting both matter and gauge fields as physical).

Most of our models, except the preceding one, were special under this gauging procedure as they allowed for no local gauge fluctuations terms and exhibited a self-duality between the topological and trivial phases. As we will show, in 3D with symmetries defined by two 1D CA, gauge fluctuations are allowed (these are the $\hat{B}_{ijk}$ operators we found in Eq 67) and there is a phase in which these models exhibit fracton *topological order*. They may be thought of as the simplest fractal *symmetry enriched* topological (FSET) phases (this possibility was already hinted at in Ref 36). The phenomenology of the resulting topological orders are the same as those of the Yoshida fractal codes [32]. The $\mathbb{Z}_2^{(a)}$ symmetry will serve the purpose of the enriching symmetry, while the other symmetries will have the interpretation of being logical operators for the underlying Yoshida code.

To avoid complications, let specialize to an $L \times L \times L$ 3-torus with $f_1^L = f_2^L = 1$ ($x^L = y^L = z^L = 1$). The symmetries in this case are given by Eq 66 and 68, but with $F_1 = \sum_{l=0}^{L}(f_1 y)^l$ and $F_2 = \sum_{l=0}^{L}(f_2 z)^l$ instead of $\mathscr{F}_1$, $\mathscr{F}_2$, with $q$ still arbitrary. There are $L$ independent $\mathbb{Z}_2^{(a)}$ symmetries, and $2L$ independent higher-form $\mathbb{Z}_2^{(b/c)}$ symmetries. An independent basis for these symmetries are, for $\alpha = 0 \ldots L-1$, given by

$$S_\alpha^{(a)} = X \begin{pmatrix} x^\alpha F_1(x,y) F_2(x,z) \\ 0 \\ 0 \end{pmatrix} \tag{69}$$

and

$$S_\alpha^{(b)} = Z \begin{pmatrix} 0 \\ x^\alpha \bar{F}_1(x,y) \\ 0 \end{pmatrix}; \qquad S_\alpha^{(c)} = Z \begin{pmatrix} 0 \\ 0 \\ x^\alpha \bar{F}_2(x,z) \end{pmatrix}. \tag{70}$$

All remaining symmetry elements may be written as products of these and $\hat{B}_{ijk}$ (as $S^{(b/c)}$ are higher-form fractal symmetries).

The fracton topologically ordered phase corresponds to the limit in which we take $h_x$ in Eq 65 to be large. Expanding about this limit, the Hamiltonian looks like

$$\mathscr{H}_{\text{FSET}} = -h_x \sum_{ijk} X \begin{pmatrix} 1 \\ 0 \\ 0 \end{pmatrix} - G \sum_{ijk} X \begin{pmatrix} 1 \\ \alpha \\ \beta \end{pmatrix} - K \sum_{ijk} Z \begin{pmatrix} 0 \\ \bar{\beta} \\ \bar{\alpha} \end{pmatrix} + (\text{perturbations}), \tag{71}$$

where we have now specified an energy scale $G$ for the second term, the third term is the leading order perturbative correction to the Hamiltonian, and we neglect all the other perturbations. Fixing all $\hat{X}_{ij}^{(a)} = 1$ results in exactly the Yoshida code

$$\mathscr{H}_{\text{Yoshida}} = -\sum_{ijk} X \begin{pmatrix} \alpha \\ \beta \end{pmatrix} - \sum_{ijk} Z \begin{pmatrix} \bar{\beta} \\ \bar{\alpha} \end{pmatrix}, \tag{72}$$

which exhibits a ground state degeneracy (with our geometry and choice of $f_{1/2}$) of $2^k$ with $k = 2L$.

From the perspective of the original FSPT, one finds that the charge of all the $S_\alpha^{(a/b)}$ (Eq 70) in the ground state of this phase no longer has to be $+1$, but instead may be $\pm 1$. These are exactly the logical operators of the Yoshida fractal code [32]. This transition may also be thought of as some kind of non-local spontaneous symmetry breaking of the higher form fractal symmetries $\mathbb{Z}_2^{(b/c)}$.

The ground state must still be uncharged under the $\mathbb{Z}_2^{(a)}$. We define the fracton excitation as an excitation of only the first term in the $\mathscr{H}_{\text{FSET}}$ (these are the relevant charge excitations when $G$ is large). Such an excitation may be created in multiplets by (for example) an operator of the form

$$Z \begin{pmatrix} 1 + (\bar{f}_1 \bar{y})^\ell \\ 1 + (\bar{f}_1 \bar{y}) + (\bar{f}_1 \bar{y})^2 + \cdots + (\bar{f}_1 \bar{y})^{\ell-1} \\ 0 \end{pmatrix}, \tag{73}$$

which creates only excitations of the first term at locations given by the non-zero coefficients in $1 + (\bar{f}_1 \bar{y})^\ell$, and is a few-body creation operator whenever $\ell = 2^l$. A single such excitation clearly carries charge $-1$ under some $\mathbb{Z}_2^{(a)}$ symmetries. This Hamiltonian therefore describes a fracton topologically ordered phase, enriched by an additional $\mathbb{Z}_2^{(a)}$ symmetry, and is a genuine FSET. As a helpful analogy, in Appendix A we show how, in exactly the same way, a relaxed

Ising gauge theory may be interpreted as an SPT protected by a global $\mathbb{Z}_2$ and 1-form $\mathbb{Z}_2$ symmetry, and in a certain limit describe an SET phase enriched by a global $\mathbb{Z}_2$.

A single charge is immobile, as discussed in Sec 5.4, provided that $f_1$ and $f_2$ are not algebraically related, the same condition which implies the lack of a string-like logical operator in the Yoshida code [32]. Finally, we note that Haah's cubic code [33] is isomorphic to this type of model, but with a second-order CA along one time direction [32].

# 7 Conclusion

We have constructed and characterized a family of Ising Hamiltonians that are symmetric under symmetry operations which involve acting on a fractal subset of spins. Fractal structures on a lattice are taken to be those defined by cellular automata with linear update rules. We discuss some possible phases in systems with such symmetries.

These include the trivial symmetric and spontaneously symmetry broken phases which are symmetric under the fractal symmetry. These fractal symmetries form the total symmetry group $(\mathbb{Z}_2)^k$, where $k$ will depend strongly on system size and topology. We then construct non-trivial symmetric phases, FSPT phases, via a decorated defect approach. For fractal symmetry groups generated by a single CA, the decorated defect construction leads to a family of cluster type Hamiltonians which have a non-trivial gapped ground state under the symmetry group $(\mathbb{Z}_2 \times \mathbb{Z}_2)^k$ of fractal symmetries. We characterize such a phase by means of symmetry-twisting, ungappable edge modes, and immobile excitations protected by the set of all fractal symmetries.

In three dimensions, our construction leads to an FSPT protected by a combination of the usual fractal symmetry along with a higher form fractal symmetry. Aside from the FSPT phase one also has the possibility of fracton topological order, enriched by the fractal $\mathbb{Z}_2$ symmetries. The topological order in these models are those of the Yoshida fractal codes [32]. While maintaining our fractal symmetries, these topologically ordered phases may be thought of as simple fractal symmetry enriched topological phases (FSET), in which an elementary excitation is charged under the fractal symmetries.

This construction is may also be generalized to higher $D$-dimensional systems, where one may consider fractal symmetries generated by $n$ $d$-dimensional CA, with $D = n + d$. The $D = 3$ dimensional models examined in this paper have $(n, d) = (1, 2)$ and $(2, 1)$. This suggests an avenue towards constructing higher dimensional fractal SPT or topological phases. Finally, we note that a generalization to $p$-state Potts variables, rather than Ising, is also possible.

Recent work in Ref [107] develops a gauging/ungauging procedure for quantum error-correcting codes. They provide a prescription for obtaining a $D$ dimensional SPT from a gapped domain wall of a $D + 1$ dimensional quantum code. The example provided of a 2D FSPT obtained from this method is exactly isomorphic to our Sierpinski FSPT.

## Acknowledgements

T.D. thanks Michael Zaletel and Frank Pollmann for inspiration and useful discussions. S.L.S. thanks Abhinav Prem for telling him about the Newman-Moore model.

**Funding information**   T.D. is funded by the DOE SciDAC program, FWP 100368 DE-AC02-76SF00515. FJB is grateful for the financial support of NSF DMR-1352271 and the Sloan Foundation FG-2015-65927.

## A Relaxed Ising gauge theory as SPT and SET phases

In this appendix, we show how the symmetry enrichment of the FSET in Sec 6.2.1 works for the simplest case: that of the $\mathbb{Z}_2$ topological order enriched by a global symmetry. We start with the Ising model on a square lattice,

$$\mathcal{H}_{\text{Ising}} = -\sum_{\langle i,j \rangle} \tau_i^z \tau_j^z, \tag{74}$$

where $i$ and $j$ label sites, and the sum is over nearest neighbors $\langle i,j \rangle$, and $\tau_i^{x/y/z}$ are Pauli matrices on site $i$. We gauged this by introducing gauge fields $\sigma_{ij}^{x/y/z}$ on every bond $ij$ between sites $i$ and $j$, and writing

$$\mathcal{H} = -\sum_{\langle i,j \rangle} \tau_i^z \tau_j^z \sigma_{ij}^z, \tag{75}$$

along with the gauge constraint that on every site $G_i|\psi\rangle = +|\psi\rangle$, with

$$G_i = \tau_i^x \prod_{j \in \Gamma(i)} \sigma_{ij}^x, \tag{76}$$

where $\Gamma(i)$ is the set of all nearest neighbors of $i$.

Next, we follow the procedure of Sec 6.2.1, we *relax* the gauge constraint and enforce it only as an energetic constraint, adding it to the Hamiltonian with coefficient $G$,

$$\mathcal{H} = -\sum_{\langle i,j \rangle} \tau_i^z \tau_j^z \sigma_{ij}^z - G \sum_i \tau_i^x \prod_{j \in \Gamma(i)} \sigma_{ij}^x. \tag{77}$$

We now interpret this Hamiltonian not as a gauge theory, but as a physical model. This model has a global symmetry

$$S_{\text{global}} = \prod_i \tau_i^x, \tag{78}$$

along with the 1-form symmetries

$$S_{\mathscr{C}} = \prod_{\langle ij \rangle \in \mathscr{C}} \sigma_{ij}^z, \tag{79}$$

where $\mathscr{C}$ is any closed loop on the square lattice. We enforce that both types of symmetries be respected, and add symmetry respecting perturbations,

$$\mathcal{H}_{\text{SPT}} = -\sum_{\langle i,j \rangle} \tau_i^z \tau_j^z \sigma_{ij}^z - G \sum_i \tau_i^x \prod_{j \in \Gamma(i)} \sigma_{ij}^x - h_x \sum_i \tau_i^x - h_z \sum_{\langle ij \rangle} \sigma_{ij}^z, \tag{80}$$

which we claim describes an SPT phase protected by the combination of the global $\mathbb{Z}_2$ and the set of 1-form symmetries. Indeed, one can verify that the edge theory must either be gapless or spontaneous symmetry breaking.

At large $h_x$, we claim this describes an SET phase. In this limit,

$$\mathcal{H}_{\text{SET}} = -h_x \sum_i \tau_i^x - G \sum_i \tau_i^x \prod_{j \in \Gamma(i)} \sigma_{ij}^x - K \sum_{\square} \prod_{\langle ij \rangle \in \square} \sigma_{ij}^z + \dots, \tag{81}$$

the ground state is clearly simply the state with all $\tau^x = 1$ and $\sigma^{x/z}$ in the ground state of the toric code.

Let us take $G$ to still be the largest energy scale. Then, the relevant charge excitations are those of the $h_x$ term, and two such excitations are created by a string of $\sigma^z$ terminated by

$\tau^z$ on either end. A single such excitation therefore carries charge $-1$ under the global $\mathbb{Z}_2$ symmetry.

To verify that this indeed describes an SET, we may gauge the global $\mathbb{Z}_2$ symmetry and verify that the charge excitation has non-trivial braiding statistics with the resulting gauge flux. Let us gauge the global $\mathbb{Z}_2$ symmetry (again), by introducing gauge fields $\mu_{ij}^{x/y/z}$ on each bond, along with the gauge constraint given by $\tilde{G}_i = \tau_i^x \prod_{j\in\Gamma(i)} \mu_{ij}^x$. We then allow for gauge fluctuations, in the form of a $\prod_\square \mu^z$ term. Then, we may gauge-fix out the $\tau$ and we are left with the Hamiltonian

$$\mathscr{H}_{\text{Gauged}} = -h_x \sum_i \prod_{j\in\Gamma(i)} \mu_{ij}^x - G \sum_i \prod_{j\in\Gamma(i)} \mu_i^x \prod_{j\in\Gamma(i)} \sigma_{ij}^x - K \sum_\square \prod_{\langle ij\rangle\in\square} \sigma_{ij}^z - K' \sum_\square \prod_{\langle ij\rangle\in\square} \mu_{ij}^z + \dots. \tag{82}$$

Recalling that the charge excitation is an excitation of the first term, we may create two such excitations at sites $i_0$ and $j_0$ by a double-string-like operator of the form

$$F(i_0, j_0) = \prod_{\langle ij\rangle\in\mathscr{C}_{i_0 j_0}} \mu_{ij}^z \sigma_{ij}^z, \tag{83}$$

where $\mathscr{C}_{i_0 j_0}$ is a path terminating at sites $i_0, j_0$. One can clearly see that moving this charge around in a closed loop measures both the original flux $\prod_\square \sigma^z$ inside the loop, but also the gauge flux $\prod_\square \mu^z$ inside it. This charge excitation therefore has mutual braiding statistics with both the original flux (as expected) but also the new gauge flux of the global $\mathbb{Z}_2$. Hence, $\mathscr{H}_{\text{SET}}$ describes a genuine SET phase, albeit very simple one.

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
