# Peer review of "Fractal Symmetric Phases of Matter"

_SciPost Physics, doi:SciPost Phys. 6, 007 (2019)_

## Round 2 · Referee Report · Anonymous · 2018-9-7

Strengths
This paper has imported many important ideas in symmetry-protected topological order with global symmetry to systems with subsystem/fractal symmetries.
Weaknesses
I can believe most of the arguments, but those on protected edge modes are not at a similar level of rigor as the other parts of the paper. See Requested changes.
Report
This paper studies quantum phase of matter that has a symmetry group generated by operators on a fractal subsystem. Important ideas from SPT literature such as robust edge modes, symmetry twist, and duality (gauging) are imported to this setting, and combined with certain fundamental immobility of excitations due to fractal symmetries. I believe this combination is a nontrivial contribution to literature.
The language of polynomials they use may seem unusual, but is somewhat inevitable since the underlying fractals are generated by linear cellular automata where polynomials are prevalent. I appreciate the authors' effort to remain in conventional language as much as possible.
Requested changes
* Among many sections of good exposition, there is one section that comes short. In Sec. 5.3, the basis of the argument is the nontriviality of the projective representation induced at the edge by the two symmetry operators that commute in the bulk. This basis alone is solid; in 1D SPT with a global internal symmetry, one can consider manifestly commuting two subgroups of the full symmetry group, inspect the induced symmetry action on the edge, and find that those at one edge is nontrivially projective. Here, it is crucial that two subgroups manifestly commuted in the bulk. I don't think this is the case for the symmetry described in page 15. The anticommuting pair of factors of the symmetry operator might be compensated by some near-edge tensor factors, and it is not clear, at least not explained in the manuscript, whether such near-edge compensation is irrelevant. Without this step, the discussion of projective representation is ill-founded.
* Some minor suggestions:
* The word symmetry is used for two technically different things. In the manuscript, sometimes it means the symmetry group, and sometimes it is an element of the symmetry group, and sometimes it means generator of some subgroup of the symmetry group. I managed to figure out which means which from the context, but it would read better if carefully and technically the three were distinguished. There are some standard way of saying it. Instead of "number of symmetries", one typically speak of the order of the symmetry group, for example.
* Around Eq.(23) the correlation function decays at a rate of volume of the fractal object (with Hausdorff dimension) but lacks any further explanation. Why is it the case?
* Below Eq.(26) it is not controlled-Z but is contolled-X in the present basis.
* "equal superposition" => equal amplitude superposition.
* On page 15, the second to the last paragraph, "faithful representation" is a technical and well-established term in the representation theory, but the usage here is improper. The proper wording is "linear representation", or one could simply say "usual representation." The faithfulness means the injectivity from the group under consideration into a matrix group where the former is represented into.
Author: Trithep Devakul on 2018-10-05 [id 326]
(in reply to Report 1 on 2018-09-07)We thank the referee for the careful reading of our manuscript and for the positive and detailed report.
One of the main requested changes is regarding Sec. 5.3 regarding the edge modes, copied below:
In response, in our edited version we have significantly modified Sec 5.3 to include more details on the edge modes and the action of the symmetries in terms of these edge modes. We believe this makes it clear exactly how each symmetry acts in terms of edge modes. The symmetries chosen in page 15 are chosen such that they act only as a single Pauli on a particular edge, trivially in the bulk, and as non-trivial Paulis on other edges, and so there are no further near-edge factors. We believe our modifications to this section make this point clear.
Responses to further suggestions:
Some minor suggestions:
The word symmetry is used for two technically different things. In the manuscript, sometimes it means the symmetry group, and sometimes it is an element of the symmetry group, and sometimes it means generator of some subgroup of the symmetry group. I managed to figure out which means which from the context, but it would read better if carefully and technically the three were distinguished. There are some standard way of saying it. Instead of "number of symmetries", one typically speak of the order of the symmetry group, for example.
We thank the referee for this valuable comment and have integrated this into the next version.
We have added an explanation of this where it is mentioned. After the duality mapping into defect variables, the Newman-Moore model (ignoring boundary conditions) becomes a non-interaction paramagnet. The 3-body correlation function is mapped to a many-body correlation function of all defect variables on the volume of the Sierpinski gasket, which therefore decays according to the volume.
Thanks for this correction.
We have implemented this change.
Indeed, the intended meaning here was "linear representation". We have corrected this accordingly.

---

## Round 3 · Referee Report · Anonymous (Referee 1) · 2018-12-7

Report

My previous concern was that the argument for the protected edges was insufficient. The present revision contains satisfactory explanation on how to find symmetry operators that "localizes" on edges, so as to induce a projective representation on one of the edges. I believe this manuscript is suitable for publication.

---

## Round 3 · Author Response

We have implemented a number of changes based on the useful comments by the referee.

---

## Round 3 · List of Changes

The changes are in response to the referee comments, which include:

  • The edge modes in Sec 5.3 are worked out in more detail.
  • Added an explanation for the correlation function of the Newman-Moore model.
  • Some wording changes through to distinguish between various concepts such as the total symmetry group as opposed to a particular element of the group, and so on.
  • Other small changes in wording as suggested by the referee.

---

## Editorial Decision

published